# Dorsolateral prefrontal activity supports a cognitive space organization of cognitive control

**Guochun Yang**[1,2,3,4], **Haiyan Wu**[5], **Qi Li**[6], **Xun Liu**[1,2]*, **Zhongzheng Fu**[7], **Jiefeng Jiang**[3,4]

[1]CAS Key Laboratory of Behavioral Science, Institute of Psychology, Beijing, China; [2]Department of Psychology, University of Chinese Academy of Sciences, Beijing, China; [3]Department of Psychological and Brain Sciences, University of Iowa, Iowa City, United States; [4]Cognitive Control Collaborative, University of Iowa, Iowa City, United States; [5]Centre for Cognitive and Brain Sciences and Department of Psychology, University of Macau, Macau, China; [6]Beijing Key Laboratory of Learning and Cognition, School of Psychology, Capital Normal University, Beijing, China; [7]Department of Neurological Surgery, Unversity of Texas Southwestern Medical Center, Dallas, United States

*For correspondence: liux@psych.ac.cn

Competing interest: The authors declare that no competing interests exist.

**Abstract** Cognitive control resolves conflicts between task-relevant and -irrelevant information to enable goal-directed behavior. As conflicts can arise from different sources (e.g., sensory input, internal representations), how a limited set of cognitive control processes can effectively address diverse conflicts remains a major challenge. Based on the cognitive space theory, different conflicts can be parameterized and represented as distinct points in a (low-dimensional) cognitive space, which can then be resolved by a limited set of cognitive control processes working along the dimensions. It leads to a hypothesis that conflicts similar in their sources are also represented similarly in the cognitive space. We designed a task with five types of conflicts that could be conceptually parameterized. Both human performance and fMRI activity patterns in the right dorsolateral prefrontal cortex support that different types of conflicts are organized based on their similarity, thus suggesting cognitive space as a principle for representing conflicts.

## eLife assessment

Yang et al. investigate whether distinct sources of conflict are represented in a common cognitive space. The study uses an interesting task that mixes different sources of difficulty and reports that the brain appears to represent these sources as a mixture on a continuum in prefrontal areas. While the findings could be **valuable** to theory in this area, there are some concerns with the design and results that raise uncertainty regarding the main conclusion of a shared cognitive space. The authors appropriately acknowledge these limitations while also highlighting the valid contributions that the study makes. Thus, while **solid** evidence is reported here, consistent with the central hypothesis, further experiments are required to support the strictest interpretation.

## Introduction

Cognitive control enables humans to behave purposefully by modulating neural processing to resolve conflicts between task-relevant and task-irrelevant information. For example, when naming the color of the word 'BLUE' printed in red ink, we are likely to be distracted by the word meaning, because

**eLife digest** You are reading a book in your local coffeeshop, when your focus gets broken by the couple at the next table, passionately discussing mortgage rates. To minimise this interruption your brain engages in 'cognitive control', resolving conflicts between competing stimuli to prioritise one over another.

Having finally regained your focus, another distraction emerges, this time of a different nature. Does your brain use the same mental mechanisms as before, and therefore a common brain circuit? Or does each kind of stimulus require a specific process?

Tasks that involve successively presenting different distractors can help explore these questions by testing for a process known as generalization: if the same mental mechanism underpins the resolution of all conflicts, distractors should become easier to ignore after the first trial.

Based on this paradigm, Yang et al. recorded brain activity during a modified version of a spatial Stroop-Simon task. Participants were asked to press a left or right button based on whether an arrow was pointing up or down, with both the vertical and horizontal position of the symbol potentially causing interference. For instance, accurate decision-making may be impaired when an arrow 'down' the bottom of the screen is pointing up (Stroop effect); or when participants must press the left button for an arrow shown on their right (Simon effect). Overall, the arrows could appear in 10 possible locations, giving rise to five types of conflicts with a unique blend of Stroop and Simon effects, with different levels of similarity.

The results showed that the degree to which conflicts could generalize to each other depended on their similarity: the more similar the conflicts, the easier it was to resolve one after having faced another. This is contrary to previous views suggesting that different conflict types either entirely generalized or could not generalize at all.

In addition, the analyses revealed that the neural networks involved in resolving each conflict type were organised in a continuous manner within a region called the prefrontal cortex. This pattern resembles how spatial information is arranged in the brain, prompting Yang et al. to suggest that cognitive control also falls under a set of principles known as cognitive space representations.

Overall, the methodology employed in this work could prove useful to researchers from other fields who also investigate whether various stimuli are processed via the same or different neural networks.

reading a word is highly automatic in daily life. To keep our attention on the color, we need to mobilize the cognitive control processes to resolve the conflict between the color and word by boosting/suppressing the processing of color/word meaning. As task-relevant and task-irrelevant information can come from different sources, the sources of conflicts and how they should be resolved can vary greatly (*Kornblum et al., 1990*). For example, the conflict may occur between items of sensory information, such as between a red light and a police officer signaling cars to pass. Alternatively, conflict may occur between sensory and motor information, such as when a voice on the left asks you to turn right. A key unsolved question in cognitive control is how our brain efficiently resolves these different types of conflicts.

A first step to addressing this question is to examine the commonalities and/or dissociations across different types of conflicts that can be categorized into different *domains*. Examples of the domains of conflicts include experimental paradigm (*Freitas et al., 2007*; *Magen and Cohen, 2007*), sensory modality (*Hazeltine et al., 2011*; *Yang et al., 2017*), or conflict type regarding the dimensional overlap of conflict processes (*Jiang and Egner, 2014*; *Liu et al., 2004*).

Two solutions to resolving different conflict types are proposed. They differ based on whether the same cognitive control mechanisms are applied across domains. On the one hand, the *domain-general* cognitive control theories posit that the frontoparietal cortex adaptively encodes task information and can thus flexibly implement control strategies for different types of conflicts. This is supported by the generalizable control adjustment (i.e., encountering a conflict trial from one type can facilitate conflict resolution of another type) (*Freitas et al., 2007*; *Kan et al., 2013*) and similar neural patterns (*Peterson et al., 2002*; *Wu et al., 2020*) across distinct conflict tasks. A broader domain-general view holds that the frontoparietal brain regions/networks are widely involved in multiple control demands well beyond the conflict domain (*Assem et al., 2020*; *Cole et al., 2013*), which explains the

remarkable flexibility in human behaviors. However, since domain-general processes are by definition likely shared by different tasks, when we need to handle multiple task demands at the same time, the efficiency of both tasks would be impaired due to resource competition or interference (*Musslick and Cohen, 2021*). Therefore, the domain-general processes are evolutionarily less advantageous for humans to deal with the diverse situations requiring high efficiency (*Cosmides and Tooby, 1994*). On the other hand, the *domain-specific* theories argue that different types of conflicts are handled by distinct cognitive control processes (e.g., where and how information processing should be modulated) (*Egner, 2008*; *Kim et al., 2012*). However, according to the domain-specific view, the diverse conflict situations require a multitude of preexisting control processes, which is biologically implausible (*Abrahamse et al., 2016*).

To reconcile the two theories, researchers recently proposed that cognitive control might be a mixture of domain-general and domain-specific processes. For instance, *Freitas and Clark, 2015* found that trial-by-trial adjustment of control can generalize across two conflict domains to different degrees, leading to domain-general (strong generalization) or domain-specific (weak or no generalization) conclusions depending on the task settings of the consecutive conflicts. Similarly, different brain networks may show domain-generality (i.e., representing multiple conflicts) or domain-specificity (i.e., representing individual conflicts separately) (*Jiang and Egner, 2014*; *Li et al., 2017*). Even within the same brain area (e.g., medial frontal cortex), *Fu et al., 2022* found that the neural population activity can be factorized into orthogonal dimensions encoding both domain-general and domain-specific conflict information, which can be selectively read out by downstream brain regions. While the mixture view provides an explanation for the contradictory findings (*Braem et al., 2014*), it suffers the same criticism as domain-specific cognitive control theories as it still requires many cognitive control processes to fully cover all possible conflicts.

A key to reconciling domain-general and domain-specific cognitive control is to organize the large number of conflict types using a system with limited, dissociable dimensions. A construct with a similar function is the *cognitive space* (*Bellmund et al., 2018*), which extends the idea of cognitive map (*Behrens et al., 2018*) to the representation of abstract information. Critically, the cognitive space view holds that the representations of different abstract information are organized continuously and the representational geometry in the cognitive space is determined by the similarity among the represented information (*Bellmund et al., 2018*).

In the human brain, it has been shown that abstract (*Behrens et al., 2018*; *Schuck et al., 2016*) and social (*Park et al., 2020*) information can be represented in a cognitive space. For example, social hierarchies with two independent scores (e.g., popularity and competence) can be represented in a 2D cognitive space (one dimension for each score), such that each social item can be located by its score in the two dimensions (*Park et al., 2020*). In the field of cognitive control, recent studies have begun to conceptualize different control states within a cognitive space (*Badre et al., 2021*). For example, *Fu et al., 2022* mapped different conflict conditions to locations in a low-/high-dimensional cognitive space to demonstrate the domain-general/domain-specific problems; *Grahek et al., 2022* used a cognitive space model of cognitive control settings to explain behavioral changes in the speed-accuracy tradeoff. However, the cognitive spaces proposed in these studies were only applicable to a limited number of control states involved in their designs. Therefore, it remains unclear whether there is a cognitive space that can explain the large number of control states, similar to that of the spatial location (*Bellmund et al., 2018*) and non-spatial knowledge (*Behrens et al., 2018*). A challenge to answering this question lies in how to construct control states with continuous levels of similarity. Our recent work (*Yang et al., 2021*) showed that it is possible to manipulate continuous conflict similarity by using a mixture of two independent conflict types with varying ratios, which can be used to further examine the behavioral and neural evidence for the cognitive space view. It is also unclear how the cognitive space of cognitive control is encoded in the brain, although that of spatial locations and non-spatial abstract knowledge has been relatively well investigated in the medial temporal lobe, medial prefrontal, and orbitofrontal system (*Behrens et al., 2018*; *Bellmund et al., 2018*). Recent research has suggested that the abstract task structure could be encoded and implemented by the frontoparietal network (*Vaidya and Badre, 2022*; *Vaidya et al., 2021*), but whether a similar neural system encodes the cognitive space of cognitive control remains untested.

The present study aimed to test the geometry of cognitive space in conflict representation. Specifically, we hypothesize that different types of conflicts are represented as points in a cognitive space.

Importantly, the distance between the points, which reflects the geometry of the cognitive space, scales with the difference in the sources of the conflicts being represented by the points. The dimensions in the cognitive space of conflicts can be the aforementioned *domains*, in which domain-specific cognitive control processes are defined. For a specific type of conflict, its location in the cognitive space can be parameterized using a limited number of coordinates, which reflect how much control is needed for each of the domain-specific cognitive control processes. The cognitive space can also represent different types of conflicts with low dimensionality (*Badre et al., 2021*; *MacDowell et al., 2022*). Different domains can be represented conjunctively in a single cognitive space to achieve domain-general cognitive control as conflicts from different sources can be resolved using the same set of cognitive control processes. We further hypothesize that the cognitive space representing different types of conflicts may be located in the frontoparietal network due to its essential roles in

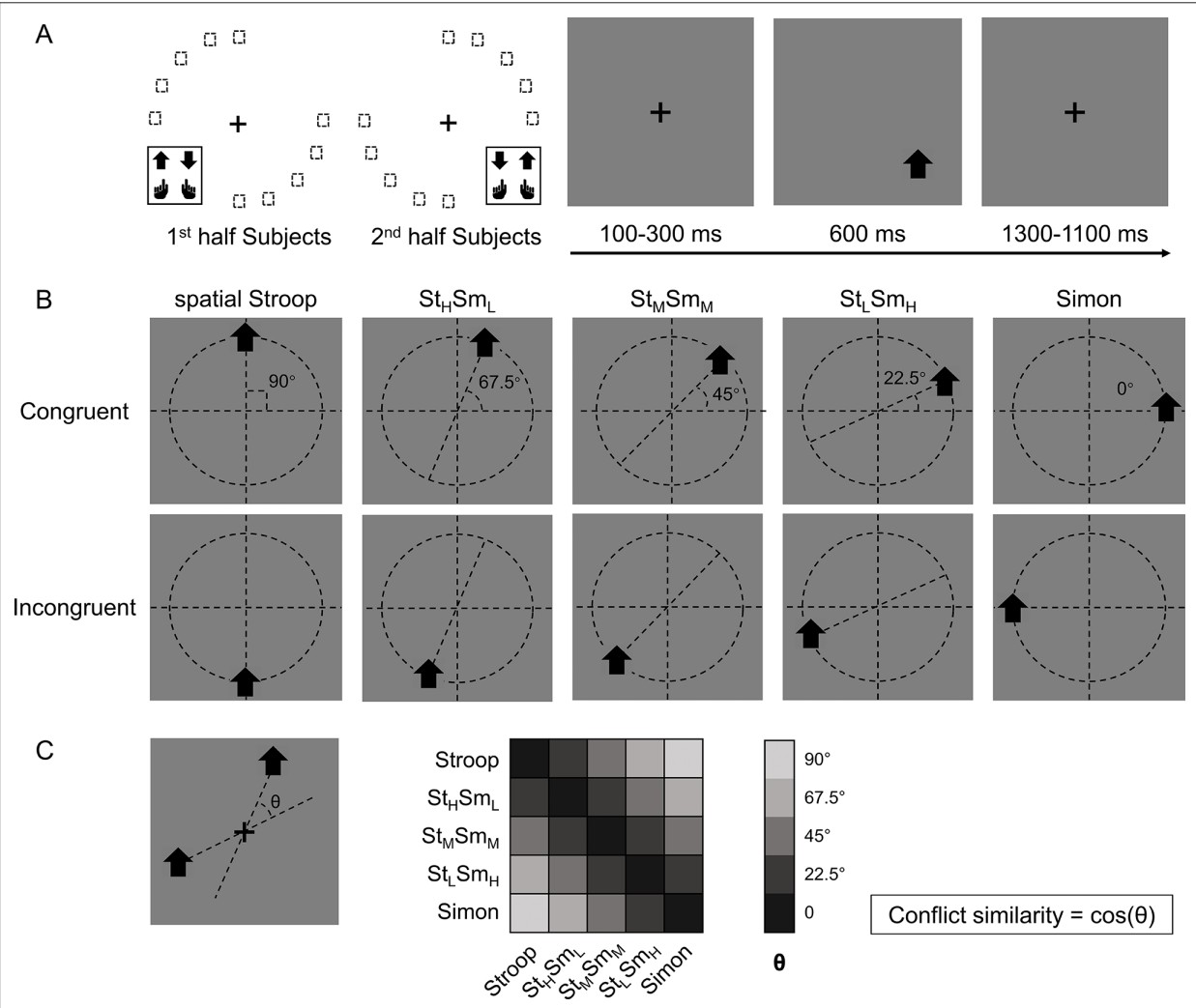

**Figure 1.** Experimental design. (**A**) The left panel shows the orthogonal stimulus–response mappings of the two participant groups. In each group, the stimuli were only displayed at two quadrants of the circular locations. One group was asked to respond with the left button to the upward arrow and with the right button to the downward arrow presented in the top-left and bottom-right quadrants, and the other group vice versa. The right panel shows the time course of one example trial. The stimuli were displayed for 600 ms, preceded and followed by fixation crosses that lasted for 1400 ms in total. (**B**) Examples of the five types of conflicts, each containing congruent and incongruent conditions. The arrows were presented at locations along five orientations with isometric polar angles, in which the vertical location introduces the spatial Stroop conflict, and the horizontal location introduces the Simon conflict. Dashed lines are shown only to indicate the location of arrows and were not shown in the experiments. (**C**) The definition of the angular difference between two conflict types and the conflict similarity. The angle $\theta$ is determined by the acute angle between two lines that cross the stimuli and the central fixation. Therefore, stimuli of the same conflict type form the smallest angle of 0, and stimuli between Stroop and Simon form the largest angle of 90°, and others are in between. Conflict similarity is defined by the cosine value of $\theta$. H = high; L = low; M = medium.

conflict resolution (*Freund et al., 2021a*; *Fu et al., 2022*) and abstract task representation (*Vaidya and Badre, 2022*).

In this study, we adjusted the paradigm from our previous study (*Yang et al., 2021*) by including transitions of trials from five different conflict types, which enabled us to test whether these conflict types are organized in a cognitive space (*Figure 1A*). Specifically, on each trial, an arrow, pointing either upward or downward, was presented on one of the 10 possible locations on the screen. Participants were required to respond to the pointing direction of the arrow (up or down) by pressing either the left or right key. Importantly, conflicts from two sources can occur in this task. On one hand, the vertical location of the arrow can be incongruent with the direction (e.g., an up-pointing arrow on the lower half of the screen), resulting spatial Stroop conflict (*Liu et al., 2004*; *Lu and Proctor, 1995*). On the other hand, the horizontal location of the arrow can be incongruent with the response key (e.g., an arrow requiring left response presented on the right side of the screen), thus causing Simon conflict (*Lu and Proctor, 1995*; *Simon and Small, 1969*). As the arrow location rotates from the horizontal axis to the vertical axis, spatial Stroop conflict increases, and Simon conflict decreases. Therefore, the 10 possible locations of the arrow give rise to five conflict types with unique blend of spatial Stroop and Simon conflicts (*Yang et al., 2021*). As the increase in spatial Stroop conflict is highly correlated with the decrease in Simon conflict, we can use a 1D cognitive space to represent all five conflict types.

One way to parameterize (i.e., defining a coordinate system) the cognitive space is to encode each conflict type by the angle of the axis connecting its two possible stimulus locations (*Figure 1B*). Within this cognitive space, the similarity between two conflict types can be quantified as the cosine value of their angular difference (*Figure 1C*). The rationale behind defining conflict similarity based on combinations of different conflict sources, such as spatial-Stroop and Simon, stems from the evidence that these sources undergo independent processing (*Egner, 2008*; *Li et al., 2014*; *Liu et al., 2010*; *Wang et al., 2014*). Identifying these distinct sources is critical in efficiently resolving diverse conflicts. If the conflict types are organized as a cognitive space in the brain, the similarity between conflict types in the cognitive space should be reflected in both the behavior and similarity in the neural representations of conflict types. Our data from two experiments using this experimental design support both predictions: using behavioral data, we found that the influence of congruency (i.e., whether the task-relevant and task-irrelevant information indicate the same response) from the previous trial to the next trial increases with the conflict similarity between the two trials. Using fMRI data, we found that more similar conflicts showed higher multivariate pattern similarity in the right dorsolateral prefrontal cortex (dlPFC).

## Results

### Behavioral congruency effects

We first tested the congruency effects for the five conflict types by conducting 5 (conflict type) × 2 (congruency) repeated-measure ANOVAs with reaction time (RT) and error rate (ER) from both experiments. The results are displayed in *Figure 2—figure supplement 1*.

### Experiment 1

For the RT, we observed a significant main effect of congruency, $F(1, 32) = 407.70$, p<0.001, $\eta_p^2 = 0.93$, a significant main effect of conflict type, $F(4, 128) = 6.32$, p<0.001, $\eta_p^2 = 0.16$, and an interaction between conflict type and congruency, $F(4, 128) = 27.86$, p<0.001, $\eta_p^2 = 0.47$. Simple effect analyses showed that participants responded more slowly in incongruent conditions than in congruent conditions for all conflict types, $p_{FDR}s<0.001$. Additionally, the congruency effects the $St_HSm_L$, $St_MSm_M$, and $St_LSm_H$ were significantly larger than that of spatial Stroop, and the congruency effects of $St_HSm_L$ and $St_MSm_M$ were significantly larger than that of Simon, $p_{FDR}s<0.05$.

Similar results were found with the ER. We observed a significant main effect of congruency, $F(1, 32) = 56.83$, p<0.001, $\eta_p^2 = 0.64$, a significant main effect of conflict type, $F(4, 128) = 6.29$, p<0.001, $\eta_p^2 = 0.16$, and an interaction between conflict type and congruency, $F(4, 128) = 13.23$, p<0.001, $\eta_p^2 = 0.29$. Simple effect analyses showed that participants were more error-prone in incongruent conditions than in congruent conditions for all conflict types, $p_{FDR}s<0.001$. The congruency effects of $St_HSm_L$, $St_MSm_M$, and $St_LSm_H$ were significantly larger than that of spatial Stroop, and the congruency effects of $St_MSm_M$ and $St_LSm_H$ were significantly larger than that of Simon, $p_{FDR}s<0.05$.

## Experiment 2

For the RT, we observed a significant main effect of congruency, $F(1, 34) = 149.71$, p<0.001, $\eta_p^2 = 0.81$, a significant main effect of conflict type, $F(4, 136) = 10.11$, p<0.001, $\eta_p^2 = 0.23$, and an interaction between conflict type and congruency, $F(4, 136) = 7.63$, p<0.001, $\eta_p^2 = 0.18$. Simple effect analyses showed that participants responded more slowly in incongruent conditions than in congruent conditions for all conflict types, $p_{FDR}s<0.001$. The congruency effect of the $St_LSm_H$ condition was larger than that of spatial Stroop, and $St_MSm_M$ and $St_LSm_H$ were significantly larger than that of Simon, $p_{FDR}s<0.05$.

For the ER, we only observed a significant main effect of congruency, $F(1, 34) = 29.80$, p<0.001, $\eta_p^2 = 0.47$. All the types showed a larger ER in incongruent than congruent conditions ($p_{FDR}s<0.001$), except that type 1 only showed a marginal significance ($p_{FDR}=0.062$).

In sum, we observed strong behavioral congruency effects in both experiments. The findings indicate that these conflict conditions indeed engaged cognitive control (*Freund et al., 2021b*).

## Conflict-type similarity modulates behavioral congruency sequence effect (CSE)

### Experiment 1

To test the influence of similarity between conflict types on behavior, we examined the CSE in consecutive trials. Specifically, the CSE was quantified as the interaction between previous and current trial congruency and can reflect how (in)congruency on the previous trial influences cognitive control on the current trial (*Egner, 2007*; *Schmidt and Weissman, 2014*). It has been shown that the CSE diminishes if the two consecutive trials have different conflict types (*Akçay and Hazeltine, 2011*; *Egner et al., 2007*; *Torres-Quesada et al., 2013*). Similarly, we tested whether the size of CSE increases as a function of conflict similarity between consecutive trials. To this end, we organized trials based on a 5 (previous trial conflict type) × 5 (current trial conflict type) × 2 (previous trial congruency) × 2 (current trial congruency) factorial design, with the first two and the last two factors capturing between-trial conflict similarity and the CSE, respectively. The cells in the 5 × 5 matrix were mapped to different similarity levels according to the angular difference between the two conflict types (*Figure 1C*). As shown in *Figure 2*, the CSE, measured in both RT and ER, scaled with conflict similarity.

To test the modulation of conflict similarity on the CSE, we constructed a linear mixed-effect model to predict RT/ER in each cell of the factorial design using a predictor encoding the interaction between the CSE and conflict similarity (see 'Materials and methods'). The results showed a significant effect of conflict similarity [RT: $\beta = 0.10 \pm 0.01$, $t(1719.7) = 15.82$, p<0.001, $\eta_p^2 = 0.120$; ER: $\beta = 0.15 \pm 0.02$, $t(204.5) = 7.84$, p<0.001, $\eta_p^2 = 0.085$, *Figure 2—figure supplement 2B/E*]. In other words, the CSE increased with the conflict similarity between two consecutive trials. The conflict similarity modulation effect remained significant after regressing out the influence of physical proximity between the stimuli of consecutive trials [RT: $\beta = 0.09 \pm 0.01$, $t(1902.4) = 13.74$, p<0.001, $\eta_p^2 = 0.025$; ER: $\beta = 0.09 \pm 0.01$, $t(249.3) = 7.66$, p<0.001, $\eta_p^2 = 0.018$]. As a control analysis, we also compared this approach to a two-stage analysis that first calculated the CSE for each previous × current trial conflict type condition and then tested the modulation of conflict similarity on the CSEs (*Yang et al., 2021*). The two-stage analysis also showed a strong effect of conflict similarity [RT: $\beta = 0.58 \pm 0.04$, $t(67.5) = 14.74$, p<0.001, $\eta_p^2 = 0.388$; ER: $\beta = 0.36 \pm 0.05$, $t(40.3) = 7.01$, p<0.001, $\eta_p^2 = 0.320$, *Figure 2—figure supplement 2A/D*]. Importantly, individual modulation effects of conflict similarity were positively correlated between the two approaches (RT: $r = 0.48$; ER: $r = 0.86$, both ps<0.003, one-tailed), indicating the consistency of the estimated conflict similarity effects across the two approaches. In the following texts, we will use the terms *conflict similarity effect* and *conflict-type effect* interchangeably.

Moreover, to test the continuity and generalizability of the similarity modulation, we conducted a leave-one-out prediction analysis. We used the behavioral data from experiment 1 for this test due to its larger amount of data than experiment 2. Specifically, we removed data from one of the five similarity levels (as illustrated by the $\theta$s in *Figure 1C*) and used the remaining data to perform the same mixed-effect model (i.e., the two-stage analysis). This yielded one pair of β-coefficients including the similarity regressor and the intercept for each subject, with which we predicted the CSE for the removed similarity level for each subject. We repeated this process for each similarity level once. The predicted results were highly correlated with the original data, with $r = 0.87$ for the RT and $r = 0.84$ for the ER, ps<0.001.

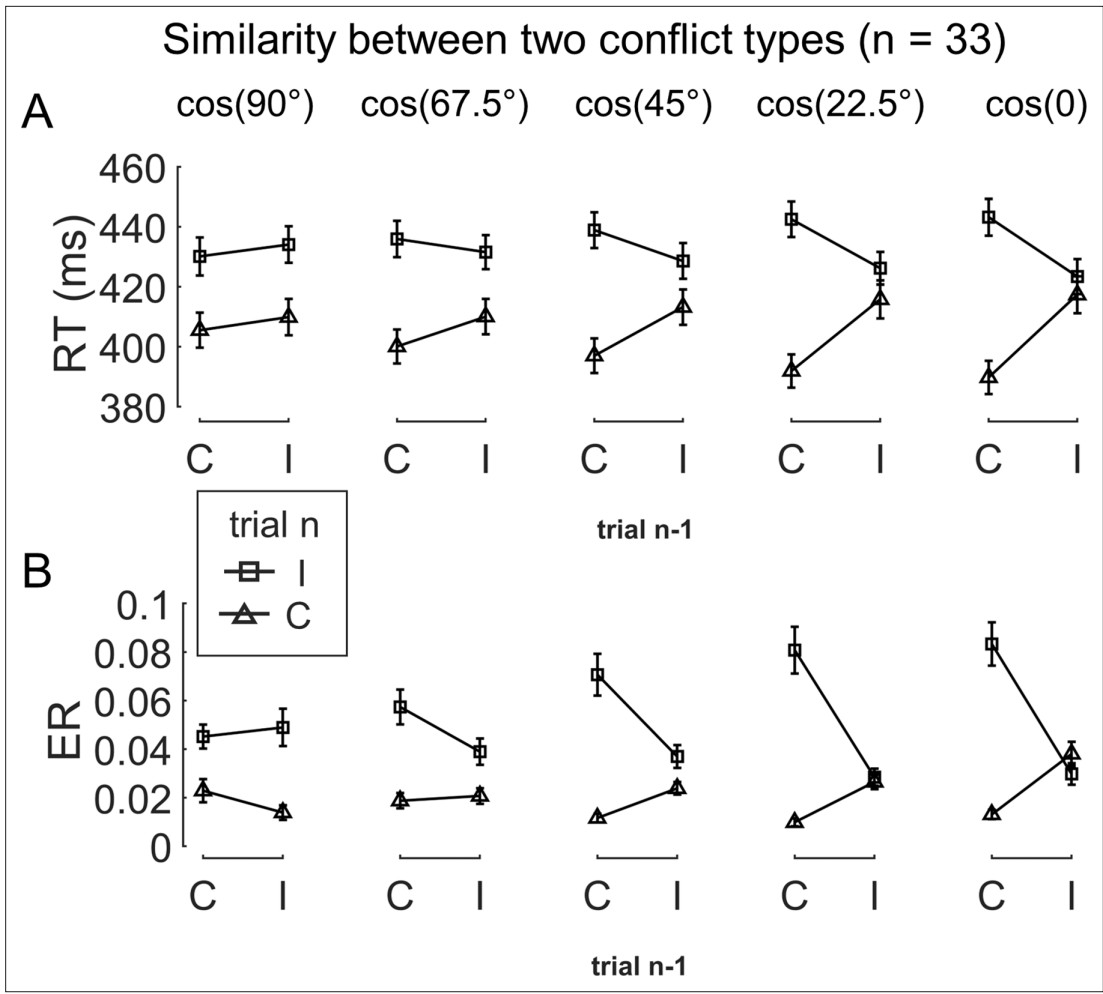

**Figure 2.** The conflict similarity modulation on the behavioral congruency sequence effect (CSE) in experiment 1. (**A**) RT and (**B**) ER are plotted as a function of congruency types on trial n−1 and trial n. Each column shows one similarity level, as indicated by the defined angular difference between two conflict types. Error bars are standard errors. C = congruent; I = incongruent; RT = reaction time; ER = error rate.

The online version of this article includes the following figure supplement(s) for figure 2:

**Figure supplement 1.** The congruency effects of experiments 1 (**A, B**) and 2 (**C, D**).

Error bars denote the standard errors of mean. Small insets on top of panel (**A**) denote an example of stimuli positions for each conflict type. RT = reaction time; ER = error rate.

**Figure supplement 2.** The conflict similarity modulation on performance of experiments 1 (**A, B, D, E**) and 2 (**C, F**), respectively.

In addition, we tested whether the conflict similarity modulation on the CSE is susceptible to training. We collected the data of experiment 1 across three sessions, thus it is possible to examine whether the conflict similarity modulation effect changes across time. To this end, we added conflict similarity, session, and their interaction into a mixed-effect linear model, in which the session was set as a categorical variable. With a post hoc ANOVA, we calculated the statistical significance of the interaction term. This approach was applied to both the RT and ER. Results showed no interaction effect in either RT, $F(2,76.4) = 1.025$, p=0.364, or ER, $F(2,49.4) = 0.789$, p=0.460. This result suggests that the modulation effect does not change across time.

## Experiment 2
### Behavioral results
We next conducted an fMRI experiment using a shorter version of the same task with a different sample (n = 35, 17 females) to seek neural evidence of how different conflict types are organized. Using behavioral data, we tested the modulation of conflict similarity on the behavioral CSE using the

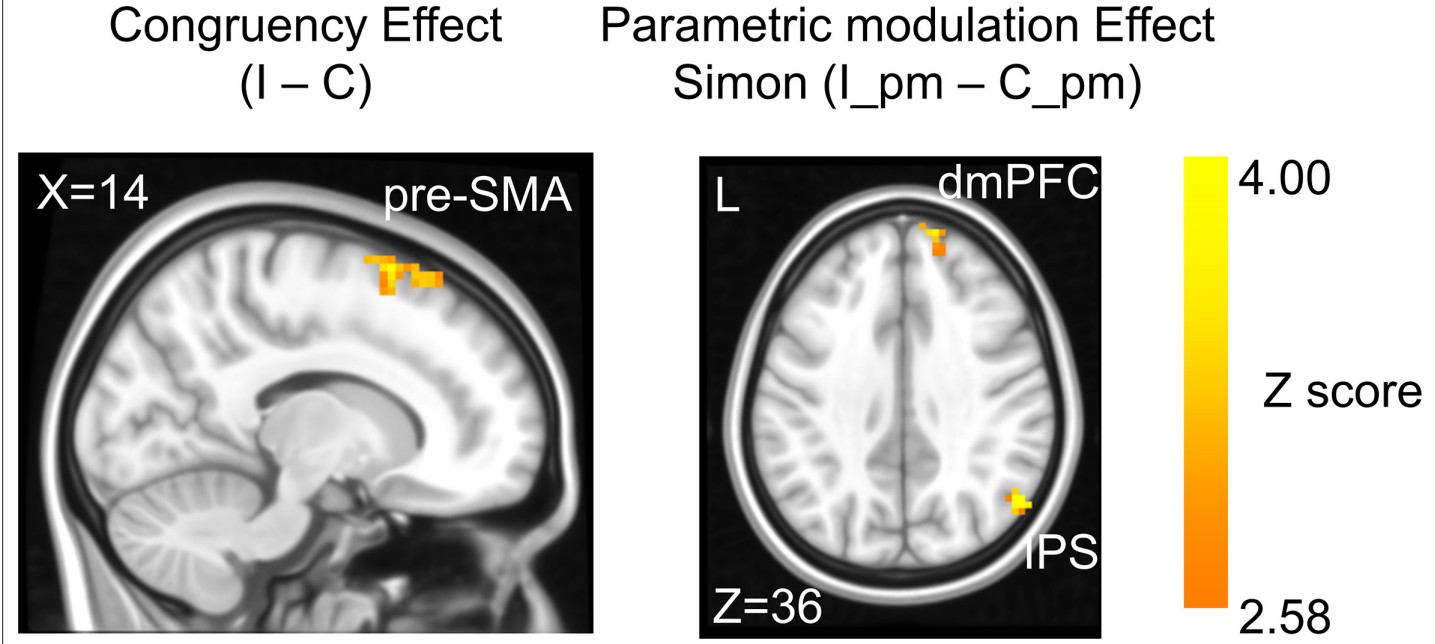

**Figure 3.** The congruency effect and parametric modulation effect detected by univoxel analyses. Results displayed are probabilistic threshold-free cluster enhancement (TFCE) enhanced and thresholded with voxel-wise p<0.001 and cluster-wise p<0.05, both one-tailed. The congruency effect denotes the higher activation in incongruent than congruent condition (left panel). The positive parametric modulation effect (I_pm – C_pm) denotes the higher activation when the conflict type contained a higher ratio of Simon conflict component (right panel). I = incongruent; C = congruent; pm = parametric modulator.

The online version of this article includes the following figure supplement(s) for figure 3:

**Figure supplement 1.** Neural congruency effect (I–C) by GLM2 (see the 'Estimation of fMRI activity with univariate general linear model'), plotted as a function of conflict type in different cortical regions of interest (ROIs).

linear mixed-effect model as in experiment 1 (except the two-stage method). Results showed a significant effect of conflict similarity modulation (RT: $\beta = 0.24 \pm 0.04$, $t(71.7) = 6.36$, p<0.001, $\eta_p^2 = .096$; ER: $\beta = 0.33 \pm 0.06$, $t(175.4) = 5.81$, p<0.001, $\eta_p^2 = 0.124$, *Figure 2—figure supplement 2C/F*), thus replicating the results of experimental 1 and setting the stage for fMRI analysis. As in experiment 1, the conflict similarity modulation effect remained significant after regressing out the influence of physical proximity between the stimuli of consecutive trials (RT: $\beta = 0.21 \pm 0.02$, $t(61.0) = 4.71$, p<0.001, $\eta_p^2 = 0.056$; ER: $\beta = 0.20 \pm 0.03$, $t(65.0) = 4.16$, p<0.001, $\eta_p^2 = 0.236$).

**Table 1.** Brain activations for the univoxel parametric analysis in GLM1 (family-wise error [FWE] corrected after probabilistic threshold-free cluster enhancement [TFCE] enhancement, with voxel-wise p<0.001 and cluster-wise p<0.05, both one-tailed).

| Region | L/R | MNI coordinate (mm) | | | Volume (no. of voxels) | MaxZ (TFCE enhanced) | BA |
|---|---|---|---|---|---|---|---|
| | | x | y | z | | | |
| *Incongruent > congruent* | | | | | | | |
| Pre-supplementary motor area | R | 12 | 12 | 73 | 71 | 4.18 | 6 |
| *Positive parametric modulator (linear Simon effect)* | | | | | | | |
| Inferior parietal sulcus | R | 52 | −64 | 33 | 81 | 4.53 | 39 |
| Dorsomedial prefrontal cortex | R | 15 | 57 | 42 | 43 | 3.92 | 9 |

L = left; R = right; BA = Brodmann area.

## Univariate brain activations scale with conflict strength

In the fMRI analysis, we first replicated the classic congruency effect by searching for brain regions showing higher univariate activation in incongruent than congruent conditions (GLM1, see 'Materials and methods'). Consistent with the literature (*Botvinick et al., 2004*; *Fu et al., 2022*), this effect was observed in the pre-supplementary motor area (pre-SMA) (*Figure 3*, *Table 1*). We then tested the encoding of conflict type as a cognitive space by identifying brain regions with activation levels parametrically covarying with the coordinates (i.e., axial angle relative to the horizontal/vertical axes) in the hypothesized cognitive space. As shown in *Figure 1B*, change in the angle corresponds to change in spatial Stroop and Simon conflicts in opposite directions, so we used opposite contrasts to examine the encoding of spatial Stroop and Simon strength, respectively (see 'Materials and methods'). Accordingly, we found the right inferior parietal sulcus (IPS) and the right dorsomedial prefrontal cortex (dmPFC) displayed positive correlation between fMRI activation and the Simon conflict (*Figure 3*, *Figure 3—figure supplement 1*, *Table 1*). We did not observe regions showing significant correlation with the spatial Stroop conflict.

To further test whether the univariate results explain the conflict similarity modulation of the behavioral CSE (slope in *Figure 2—figure supplement 2C*), we conducted brain–behavioral correlation analyses for regions identified above. Regions with higher spatial Stroop/Simon modulation effects were expected to trigger higher behavioral conflict similarity modulation effect on the CSE. However, none of the two regions (i.e., right IPS and right dmPFC, *Figure 3*) were positively correlated with the behavioral performance, both uncorrected ps>0.28, one-tailed. In addition, since the conflict-type difference covaries with the orientation of the arrow location at the individual level (e.g., the stimulus in a higher level of Simon conflict is always closer to the horizontal axis, see *Figure 4A*), the univariate modulation effects may not reflect purely conflict-type difference. To further tease these factors apart, we used multivariate analyses.

## Cross-subject representational similarity analysis (RSA) and validation

The hypothesis that the brain encodes conflict types in a cognitive space predicts that similar conflict types will have similar neural representations. To test this prediction, we computed the representational similarity matrix (RSM) that encoded correlations of blood-oxygen-level dependent (BOLD) signal patterns between each pair of conflict type (Stroop, $St_HSm_L$, $St_MSm_M$, $St_LSm_H$, and Simon, with H, M, and L indicating high, medium, and low, respectively, see also *Figure 1B*) × congruency (congruent, incongruent) × arrow direction (up, down) × run × subject combinations for each of the 360 cortical regions from the multi-modal parcellation (MMP) cortical atlas (*Glasser et al., 2016*; *Jiang et al., 2020*). The RSM was then submitted to a linear mixed-effect model as the dependent variable to test whether the representational similarity in each region was modulated by various experimental variables (e.g., conflict type, spatial orientation, stimulus, response, etc., see 'Materials and methods'). The linear mixed-effect model was used to de-correlate conflict type and spatial orientation leveraging the between-subject manipulation of stimulus locations (*Figure 4A*).

To validate this method, we applied this analysis to test the effects of response/stimulus features and found that representational similarity of the BOLD signal patterns significantly covaried with whether two response/spatial location/arrow directions were the same most strongly in bilateral motor/visual/somatosensory areas, respectively (*Figure 5*).

We further validated the cross-subject RSA by testing the hypothesis that exerting cognitive control can enhance target representation and suppress distractor representation. Our underlying assumption was that stimuli are represented in visual areas, so we chose a visual region from the main RSA results showing joint representation of target, spatial Stroop distractor, and Simon distractor (p<0.005, one-tail, uncorrected). Only the left V4 met this criterion. We then tested representations with models for incongruent-only trials, congruent-only trials, and the incongruent–congruent contrast (see 'Materials and methods'). The contrast model additionally used interaction between the congruency and target, Stroop distractor and Simon distractor terms. Results showed that in the incongruent condition, when we employ more cognitive control, the target representation was enhanced [$t(72.2) = 2.28$, p=0.039, Bonferroni corrected] and both spatial Stroop [$t(85.4) = -3.94$, p<0.001, Bonferroni corrected] and Simon [$t(38.9) = -2.80$, p=0.012, Bonferroni corrected] distractor representations were suppressed (*Figure 5E*). These are consistent with the idea that the top-down control modulates the stimuli in

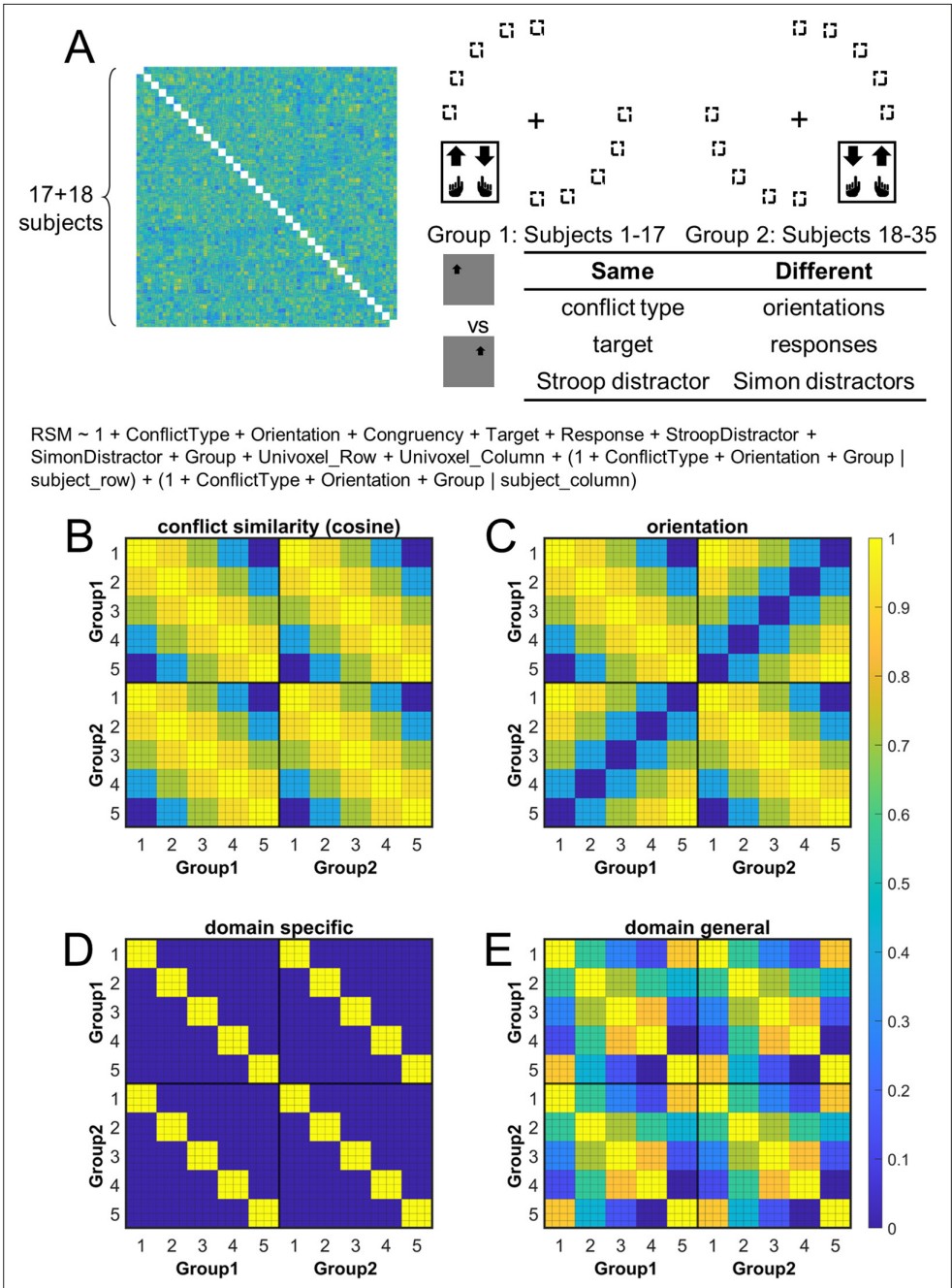

**Figure 4.** Rationale of the cross-subject representational similarity analysis (RSA) model and the schematic of key representational similarity matrix (RSM). (**A**) The RSM is calculated as the Pearson's correlation between each pair of conditions across the 35 subjects. For 17 subjects, the stimuli were displayed on the top-left and bottom-right quadrants, and they were asked to respond with left hand to the upward arrow and right hand to the downward arrow. For the other 18 subjects, the stimuli were displayed on the top-right and bottom-left quadrants, and they were asked to respond with left hand to the downward arrow and right hand to the upward arrow. Within each subject, the conflict type and orientation regressors were perfectly covaried. For instance, the same conflict type will always be on the same orientation. To de-correlate conflict type and orientation effects, we conducted the RSA across subjects from different groups. For example, the bottom-right panel highlights the example conditions that are orthogonal to each other on the orientation, response, and Simon distractor, whereas their conflict type, target, and spatial Stroop distractor are the same. The dashed boxes show the possible target locations for different conditions. (**B**) and (**C**) show the orthogonality between conflict similarity and orientation RSMs. The within-subject RSMs (e.g., group 1–group 1) for conflict similarity and orientation are all the same, but the cross-

*Figure 4 continued on next page*

*Figure 4 continued*

group correlations (e.g., group 2–group 1) are different. Therefore, we can separate the contribution of these two effects when including them as different regressors in the same linear regression model. (**D**) and (**E**) show the two alternative models. Like the cosine model (**B**), within-group trial pairs resemble between-group trial pairs in these two models. The domain-specific model is an identity matrix. The domain-general model is estimated from the absolute difference of behavioral congruency effect, but scaled to 0 (lowest similarity) – 1 (highest similarity) to aid comparison. The plotted matrices in (**B–E**) include only one subject each from groups 1 and 2. Numbers 1–5 indicate the conflict-type conditions of spatial Stroop, $St_HSm_L$, $St_MSm_M$, $St_LSm_H$, and Simon, respectively. The thin lines separate four different subconditions, that is, target arrow (up, down) × congruency (incongruent, congruent), within each conflict type.

both directions (*Polk et al., 2008*; *Ritz and Shenhav, 2022*), underscoring the utility of cross-subject RSA in our study.

## Multivariate patterns of the right dlPFC encode the conflict similarity

We then identified the cortical regions encoding conflict type as a cognitive space by testing whether their RSMs can be explained by the similarity between conflict types. Specifically, we applied three independent criteria: (1) the cortical regions should exhibit a statistically significant positive conflict similarity effect on the RSM; (2) the conflict similarity effect should be stronger in incongruent than congruent trials to reflect flexible adjustment of cognitive control demand when the conflict is present; and (3) the conflict similarity effect should be positively correlated with the behavioral conflict similarity modulation effect on the CSE (see 'Behavioral results' of experiment 2). The first criterion revealed several cortical regions encoding the conflict similarity, including the frontal eye field (FEF), region 1, Brodmann 8C area (a subregion of dlPFC) (*Glasser et al., 2016*), a47r, posterior inferior frontal junction (IFJp), anterior intraparietal area (AIP), temporoparietooccipital junction 3 (TPOJ3), PGi, and V3CD in the right hemisphere, and the superior frontal language (SFL) area, 23c, 24dd, 7Am, p32pr, 6r, FOP1, PF, ventromedial visual area (VMV1/2) areas, area 25, MBelt in the left hemisphere (Bonferroni corrected ps<0.05, one-tailed, *Figure 6A*). We next tested whether these regions were related to cognitive control by comparing the strength of conflict similarity effect between incongruent and congruent conditions (criterion 2) and correlating the strength to behavioral similarity modulation effect (criterion 3). Given these two criteria pertain to second-order analyses (interaction or individual analyses) and thus might have lower statistical power (*Blake and Gangestad, 2020*), we applied a more lenient threshold using false discovery rate (FDR) correction (*Benjamini and Hochberg, 1995*) on the abovementioned regions. Results revealed that the left SFL, left VMV1, area left 25 and right 8C met this criterion, FDR-corrected ps<0.05, one-tailed, suggesting that the representation of conflict type was strengthened when the conflict was present (e.g., *Figure 6D*, *Figure 6—figure supplement 1*). The inter-subject brain–behavioral correlation analysis (criterion 3) showed that the strength of conflict similarity effect on RSM scaled with the modulation of conflict similarity on the CSE (slope in *Figure 2—figure supplement 2C*) in right 8C (r = 0.52, FDR-corrected p=0.015, one-tailed, *Figure 6C*) only. These results are listed in *Table 2*.

We observed the right 8C but not the left 8C represented the conflict-type similarity. A further test is to show whether there is a lateralization. We tested several regions of the left dlPFC, including the i6-8, 8Av, 8C, p9-46v, 46, 9-46d, and a9-46v (*Freund et al., 2021a*). We found that none of these regions show the representation of conflict type, all uncorrected ps>0.35. These results indicate that the conflict type is specifically represented in the right dlPFC.

In addition, we repeated the analysis by using data smoothed with a 6 mm full width at half maximum (FWHM) Gaussian kernel. Results showed a significant conflict similarity effect in right 8C, $t(1902599.9) = 5.55$, p<0.0001, replicating the results on unsmoothed data [$t(86.8) = 5.41$, p<0.0001]. This observation implies that the possibility of variations in brain region locations across subjects, which may have been mitigated using a larger smoothing kernel, does not appear to have influenced the results.

The model described above employs the cosine similarity measure to define conflict similarity and will be referred to as the cognitive-space model. To examine whether the right 8C specifically encodes the cognitive space rather than the domain-general or domain-specific organizations, we tested two additional models (see 'Materials and methods'). Model comparison showed a lower Bayesian

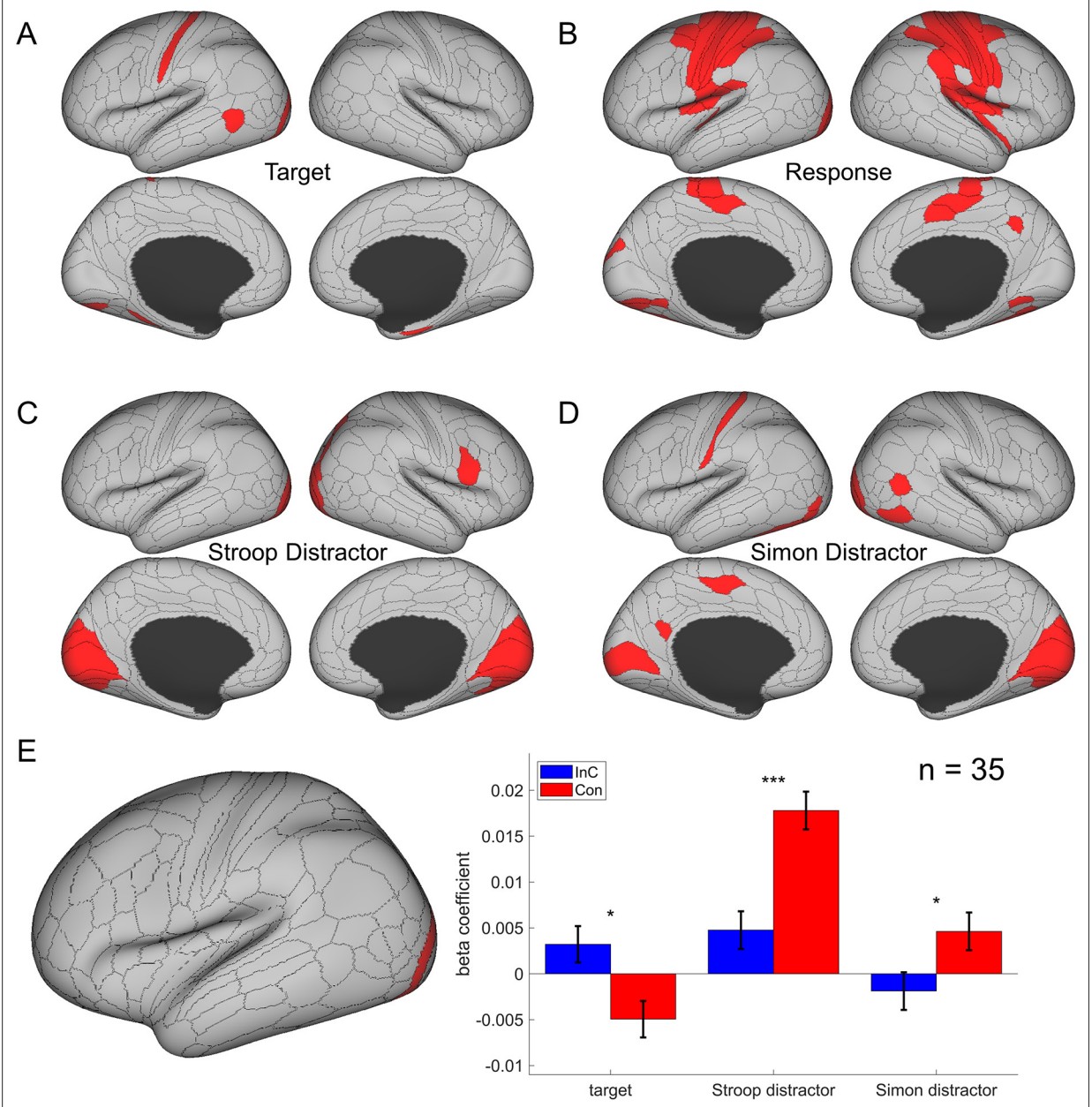

**Figure 5.** The cortical regions showing different effects in the main representational similarity analysis (RSA). (**A**) The target effect reflects the encoding of upward and downward arrow directions and is mainly encoded in the visual and sensorimotor regions. (**B**) The response effect reflects the encoding of left and right responses and is mainly encoded in motor regions. (**C**) The spatial Stroop distractor effect reflects the encoding of vertical location of the stimulus and is encoded in bilateral visual regions. (**D**) The Simon distractor effect reflects the encoding of horizontal locations of the stimulus and is mainly encoded at the bilateral visual regions. Regions in (**B–D**) are thresholded with Bonferroni-corrected p<0.05 across the 360 cortical ROIs, whereas regions in (**A**) are thresholded with uncorrected p<0.005. (**E**) The representational strength of target, Stroop distractor, and Simon distractor in left V4 (left panel) for incongruent and congruent conditions. Compared to the congruent conditions, the incongruent condition shows a stronger representation of target, but lower representation of Stroop and Simon distractors. Results are Bonferroni corrected. *p<0.05, ***p<0.001.

information criterion (BIC) in the cognitive-space model (BIC = 5,377,093) than the Domain-General (BIC = 5,377,126) or Domain-Specific (BIC = 5,377,127) models. Further analysis showed the dimensionality of the representation in the right 8C was 1.19, suggesting the cognitive space was close to 1D. Moreover, we replicated the results with only incongruent trials, considering that the pattern of conflict representations is more manifested when the conflict is present (i.e., on incongruent trials) than not (i.e., on congruent trials). We found a poorer fitting in domain-general (BIC = 1,344,127) and

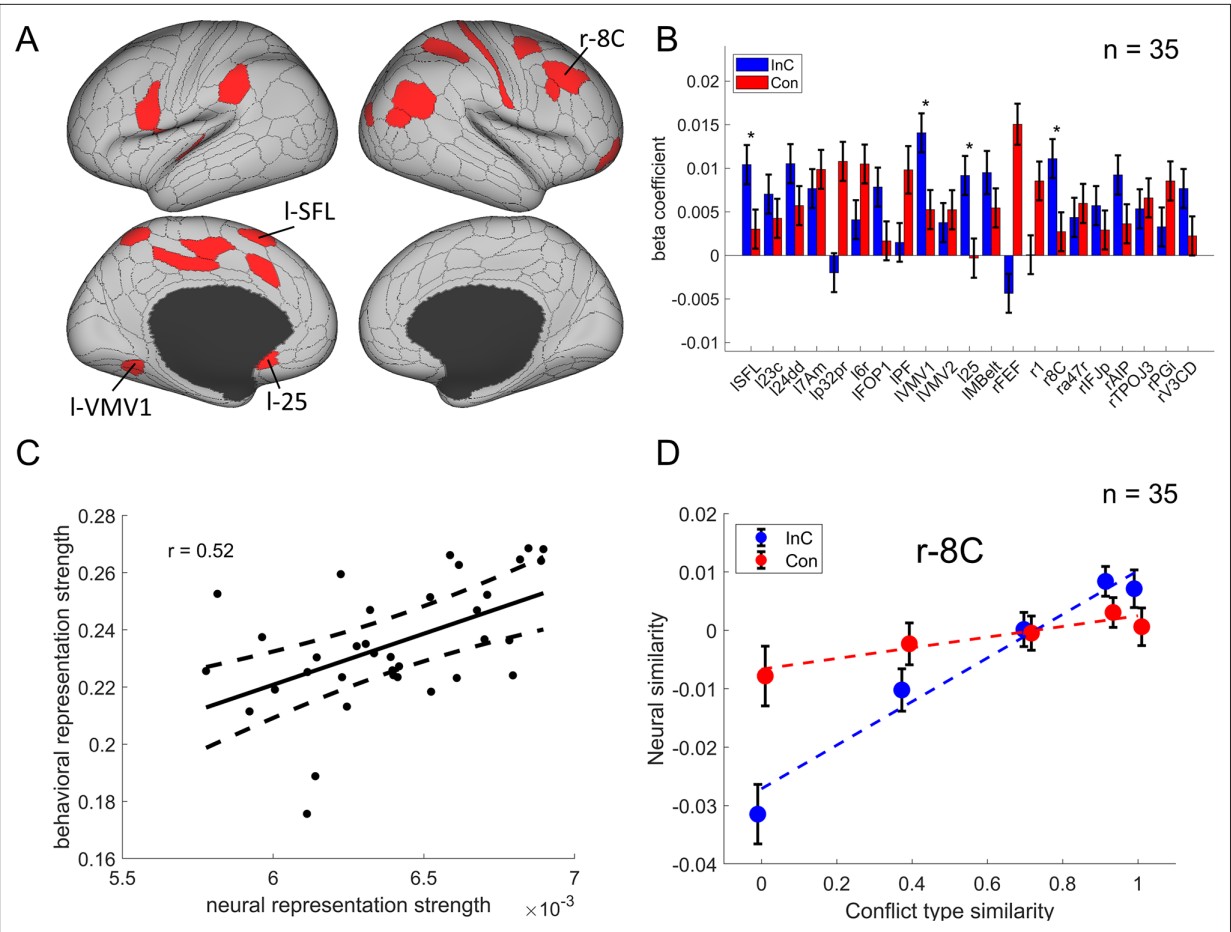

**Figure 6.** The conflict-type effect. (**A**) Brain regions surviving the Bonferroni correction (p<0.05) across the 360 regions (criterion 1). Labeled is the region meeting the three criteria. (**B**) Different encoding of conflict-type effect in the incongruent with congruent conditions (criterion 2). *False discovery rate (FDR)-corrected p<0.05. (**C**) The brain–behavior correlation of the right 8C (criterion 3). The x-axis shows the β-coefficient of the conflict-type effect from the representational similarity analysis (RSA), and the y-axis shows the β-coefficient obtained from the behavioral linear model using the conflict similarity to predict the congruency sequence effect (CSE) in experiment 2. (**D**) Illustration of the different encoding strength of conflict-type similarity in incongruent versus congruent conditions of right 8C. The y-axis is derived from the z-scored Pearson correlation coefficient after regressing out other factors. See *Figure 6—figure supplement 1B-* for a plot of similarity matrix across different conflict conditions in both incongruent and congruent conditions. l = left; r = right.

The online version of this article includes the following figure supplement(s) for figure 6:

**Figure supplement 1.** The stronger conflict-type similarity effect in incongruent versus congruent conditions.

domain-specific (BIC = 1,344,129) models than the cognitive-space model (BIC = 1,344,104). These results indicate that the right 8C encodes an integrated cognitive space for resolving Stroop and Simon conflicts. More detailed model comparison results are listed in *Table 3*.

In sum, we found converging evidence supporting that the right dlPFC (8C area) encoded conflict similarity parametrically, which further supports the hypothesis that conflict types are represented in a cognitive space.

## Multivariate patterns of visual and oculomotor areas encode stimulus orientation

To tease apart the representation of conflict type from that of perceptual information, we tested the modulation of the spatial orientations of stimulus locations on RSM using the aforementioned RSA. We also applied three independent criteria: (1) the cortical regions should exhibit a statistically significant orientation effect on the RSM; (2) the conflict similarity effect should be stronger in incongruent than congruent trials; and (3) the orientation effect should not interact with the CSE since the

**Table 2.** Summary statistics of the cross-subject representational similarity analysis (RSA) for regions showing conflict type and orientation effects identified by the three criteria.

| Region name | Criterion 1 | | | Criterion 2 | | Criterion 3 | |
|---|---|---|---|---|---|---|---|
| | t | $\beta$ ± SD | p | t | p | r | p |
| *Conflict type effect* | | | | | | | |
| Left SFL | 4.77 | 0.0061 ± 0.0013 | $1.9 \times 10^{-6}$ | 2.45 | 0.007 | 0.35 | 0.021 |
| Left 23 c | 4.42 | 0.0049 ± 0.0011 | $9.8 \times 10^{-6}$ | 1.18 | 0.118 | −0.15 | 0.800 |
| Left 24dd | 6.13 | 0.0079 ± 0.0013 | $8.9 \times 10^{-10}$ | 1.61 | 0.053 | −0.04 | 0.580 |
| Left 7Am | 6.76 | 0.0090 ± 0.0013 | $1.4 \times 10^{-11}$ | −0.75 | 0.772 | 0.04 | 0.418 |
| Left p32pr | 4.00 | 0.0044 ± 0.0011 | $6.4 \times 10^{-5}$ | −3.95 | 1.000 | −0.01 | 0.533 |
| Left 6 r | 5.41 | 0.0071 ± 0.0013 | $6.3 \times 10^{-8}$ | −1.94 | 0.973 | −0.11 | 0.737 |
| Left FOP1 | 4.37 | 0.0050 ± 0.0011 | $1.3 \times 10^{-5}$ | 1.83 | 0.034 | −0.25 | 0.926 |
| Left PF | 4.04 | 0.0058 ± 0.0014 | $5.3 \times 10^{-5}$ | −1.90 | 0.969 | −0.24 | 0.921 |
| Left VMV1 | 7.45 | 0.0091± 0.0012 | $9.6 \times 10^{-14}$ | 2.83 | 0.002 | −0.27 | 0.940 |
| Left VMV2 | 4.75 | 0.0053 ± 0.0011 | $2.0 \times 10^{-6}$ | −0.32 | 0.624 | −0.06 | 0.630 |
| Left 25 | 3.70 | 0.0041 ± 0.0011 | $2.1 \times 10^{-4}$ | 3.26 | 0.001 | 0.05 | 0.386 |
| Left Mbelt | 4.25 | 0.0064 ± 0.0015 | $2.1 \times 10^{-5}$ | 1.77 | 0.039 | 0.10 | 0.275 |
| Right FEF | 3.98 | 0.0054 ± 0.0014 | $6.8 \times 10^{-5}$ | −5.49 | 1.000 | 0.08 | 0.327 |
| Right 1 | 3.90 | 0.0045 ± 0.0012 | $9.7 \times 10^{-5}$ | −2.34 | 0.990 | −0.07 | 0.665 |
| Right 8 C* | 5.41 | 0.0064 ± 0.0012 | $6.1 \times 10^{-8}$ | 2.46 | 0.007 | 0.52 | 0.001 |
| Right a47r | 5.04 | 0.0056 ± 0.0011 | $4.7 \times 10^{-7}$ | −0.68 | 0.753 | 0.05 | 0.393 |
| Right IFJp | 3.78 | 0.0042 ± 0.0011 | $1.6 \times 10^{-4}$ | 0.77 | 0.221 | 0.27 | 0.056 |
| Right AIP | 4.64 | 0.0054 ± 0.0012 | $3.5 \times 10^{-6}$ | 1.86 | 0.032 | −0.02 | 0.540 |
| Right TPOJ3 | 4.48 | 0.0056 ± 0.0012 | $7.6 \times 10^{-6}$ | −0.25 | 0.600 | 0.21 | 0.118 |
| Right PGi | 4.07 | 0.0045 ± 0.0011 | $4.8 \times 10^{-5}$ | −1.59 | 0.944 | −0.01 | 0.523 |
| Right V3CD | 3.86 | 0.0043 ± 0.0011 | $1.2 \times 10^{-4}$ | 1.94 | 0.026 | 0.02 | 0.451 |
| *Orientation effect* | | | | | | | |
| Left FEF* | 5.17 | 0.0060 ± 0.0012 | $2.4 \times 10^{-7}$ | 2.73 | 0.003 | −0.01 | 0.518 |
| Left POS1 | 4.35 | 0.0051 ± 0.0012 | $1.4 \times 10^{-5}$ | −1.52 | 0.936 | 0.00 | 0.500 |
| Left 31pv | 5.60 | 0.0103 ± 0.0018 | $2.1 \times 10^{-8}$ | −0.48 | 0.686 | 0.05 | 0.397 |
| Left 6ma | 4.75 | 0.0055 ± 0.0012 | $2.0 \times 10^{-6}$ | −1.87 | 0.970 | −0.00 | 0.500 |
| Left 7PC | 3.66 | 0.0042 ± 0.0012 | $2.5 \times 10^{-4}$ | 0.76 | 0.223 | −0.20 | 0.876 |
| Left 8BL | 4.55 | 0.0053 ± 0.0012 | $5.3 \times 10^{-6}$ | 0.10 | 0.460 | 0.14 | 0.207 |
| Left AIP | 3.83 | 0.0044 ± 0.0012 | $1.3 \times 10^{-4}$ | 0.62 | 0.266 | −0.23 | 0.910 |
| Left TE1p | 3.95 | 0.0046 ± 0.0012 | $7.8 \times 10^{-5}$ | 0.14 | 0.443 | 0.01 | 0.475 |
| Left IP2* | 4.49 | 0.0052 ± 0.0012 | $7.2 \times 10^{-6}$ | 4.02 | 0.000 | 0.19 | 0.139 |
| Right V1* | 4.71 | 0.0083 ± 0.0018 | $2.5 \times 10^{-6}$ | 2.91 | 0.002 | −0.08 | 0.672 |
| Right V2* | 3.88 | 0.0172 ± 0.0044 | $1.1 \times 10^{-4}$ | 3.19 | 0.001 | 0.02 | 0.462 |
| Right V3 | 3.94 | 0.0175 ± 0.0045 | $8.2 \times 10^{-5}$ | −1.45 | 0.927 | 0.39 | 0.010 |
| Right LO2 | 5.95 | 0.0069 ± 0.0012 | $2.8 \times 10^{-9}$ | −1.81 | 0.965 | 0.26 | 0.068 |
| Right POS1* | 3.70 | 0.0043 ± 0.0012 | $2.1 \times 10^{-4}$ | 2.98 | 0.001 | 0.33 | 0.028 |

*Table 2 continued on next page*

*Table 2 continued*

| Region name | Criterion 1 | | | Criterion 2 | | Criterion 3 | |
|---|---|---|---|---|---|---|---|
| | t | $\beta \pm$ SD | p | t | p | r | p |
| Right 5 m | 5.24 | $0.0061 \pm 0.0012$ | $1.6 \times 10^{-7}$ | −2.12 | 0.983 | 0.00 | 0.500 |
| Right TF | 4.97 | $0.0058 \pm 0.0012$ | $6.7 \times 10^{-7}$ | −1.08 | 0.860 | −0.07 | 0.657 |
| Right PHT | 4.54 | $0.0053 \pm 0.0012$ | $5.5 \times 10^{-6}$ | 0.03 | 0.486 | −0.04 | 0.589 |
| Right PF* | 5.57 | $0.0064 \pm 0.0012$ | $2.6 \times 10^{-8}$ | 3.08 | 0.001 | −0.03 | 0.558 |
| Right A4 | 4.11 | $0.0048 \pm 0.0012$ | $3.9 \times 10^{-5}$ | −2.68 | 0.996 | −0.09 | 0.700 |

All p-values listed are one-tailed and uncorrected.
*Denotes the regions meeting all three criteria for each effect.

orientation effect was dissociated from the conflict similarity effect and was not expected to influence cognitive control. We observed increasing fMRI representational similarity between trials with more similar orientations of stimulus location in the occipital cortex, such as right V1, right V2, right V3, bilateral POS1, and right lateral occipital 2 (LO2) areas (Bonferroni corrected ps<0.05). We also found the same effect in the oculomotor related region, that is, the left FEF, and other regions including the left 31pv, 6ma, 7PC, 8BL, AIP, TE1p, IP2, right 5m, TF, PHT, A4, and parietal area F (PF) (*Figure 7A*). Then we tested whether any of these brain regions were related to the conflict representation by comparing their encoding strength between incongruent and congruent conditions. Results showed that the right V1, right V2, right POS1, left IP2, left FEF, and right PF encoded stronger orientation effect in the incongruent than the congruent condition, FDR-corrected ps<0.05, one-tailed (*Table 2*, *Figure 7B*). We then tested whether any of these regions were related to the behavioral performance, and results showed that none of them positively correlated with the behavioral conflict similarity modulation effect, all uncorrected ps>0.18, one-tailed. Thus all identified regions are consistent with criterion 3. Taken together, we found that the visual and oculomotor regions encoded orientations of stimulus location in a continuous manner and that the encoding strength was stronger when the conflict was present.

We hypothesize that the overlapping spatial information of orientation may have facilitated the encoding of conflict types. To explore the relation between conflict type and orientation representations, we conducted representational connectivity (i.e., the similarity between two within-subject RSMs of two regions) (*Kriegeskorte et al., 2008*) analyses and found that among the orientation effect regions, the right V1 and right V2 showed significant representational connectivity with the right 8C compared to the controlled regions (including those encoding orientation effect but not showing larger encoding strength in incongruent than congruent conditions, as well as eight other regions encoding none of our defined effects in the main RSA, see 'Materials and methods'). Compared with the largest connectivity strength in the controlled regions (i.e., the left 6ma, $\beta = 0.7991 \pm 0.0299$), we found higher connectivity in the right V1, $\beta = 0.8633 \pm 0.0325$, and right V2, $\beta = 0.8772 \pm 0.0335$ (*Figure 8*).

**Table 3.** Model comparison results of the right 8C.
RSM_I shows results using incongruent trials only.

| Model name | Full RSM | | | RSM_I | | |
|---|---|---|---|---|---|---|
| | t | p | BIC | t | p | BIC |
| Cognitive-space | 5.41 | $6.1 \times 10^{-8}$ | 5,377,093 | 4.97 | $3.35 \times 10^{-7}$ | 1,345,201 |
| Domain-general | 0.92 | .179 | 5,377,126 | 1.43 | 0.076 | 1,344,127 |
| Domain-specific | 0.84 | .200 | 5,377,127 | 0.28 | 0.390 | 1,344,129 |

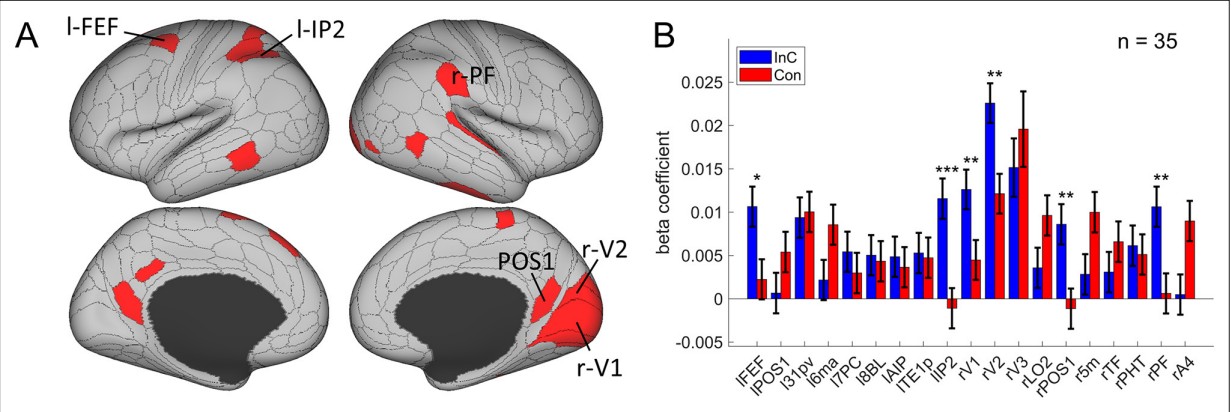

**Figure 7.** The orientation effect. (**A**) Brain regions surviving the Bonferroni correction (p<0.05) across the 360 regions (criterion 1). Labeled regions are those meeting the three criteria. (**B**) Different encoding of orientation in the incongruent with congruent conditions. *False discovery rate (FDR)-corrected p<0.05; **FDR-corrected p<0.01; ***FDR-corrected p<0.001.

## The multivariate representations of conflict type and orientation are different from the congruency effect

An explanation to the stronger encoding of conflict type in incongruent than congruent condition (*Figure 6B, D*) in right 8C area may be the encoding of congruency. To test this possibility, we first tested the univariate congruency effect (incongruent minus congruent) using the parametric modulating GLM1 that was used to estimate fMRI activation levels of conflict type × congruency conditions. We observed no univariate congruency effect in the right 8C region, $t(34) = -0.03$, p=0.513, one-tailed. Neither did we observe a multivariate congruency effect (i.e., the pattern difference between incongruent and congruent conditions compared to that within each condition) in the right 8C or any other regions. Note the definition of congruency here differed from traditional definitions (i.e., contrast between activity strength of incongruent and congruent conditions), with which we found stronger univariate activities in pre-SMA for incongruent versus congruent conditions. We further tested the possibility that the congruency effect may be manifested in behavioral relevance. To this end, we extracted the contrast of incongruent minus congruent on encoding strength of conflict similarity for each subject from the mixed-effect model based on the cross-subject RSA (see 'Representational similarity analysis') and correlated it with the behavioral congruency effect, averaged across the five conflict types (i.e., the main effect reported in *Figure 2—figure supplement 1*). No significant correlation was observed ($r = 0.14$, p=0.380, one-tailed). Taken together, these results suggested that the neural encoding strength of conflict type does not reflect the level of cognitive control engagement, but the dynamic adjustment of cognitive control instead.

Similarly, we tested whether those regions with stronger encoding of orientation in incongruent than congruent condition (i.e., right V1, V2, PF, POS1, and left FEF and IP2) reflect the congruency effect. We observed no univoxel congruency effect in any of these regions, all uncorrected ps>0.89, one-tailed. In addition, the orientation effect was not correlated to the behavioral congruency in any of the regions, all uncorrected ps>0.074, one-tailed. Together with our finding that there was no correlation between the strength of orientation encoding and the conflict similarity modulation on behavioral CSEs in any of these regions (see 'Multivariate patterns of visual and oculomotor areas encode stimulus orientation'), these results indicate that the encoding of orientation effect did not reflect the encoding of congruency or conflict type. Instead, we speculate that the encoding of orientations provides perceptual information to determine the conflict type.

## Discussion

Understanding how different types of conflicts are resolved is essential to answer how cognitive control achieves adaptive behavior. However, the dichotomy between domain-general and/or domain-specific processes presents a dilemma (*Braem et al., 2014*; *Egner, 2008*). Reconciliation of the two views also suffers from the inability to fully address how different conflicts can be resolved

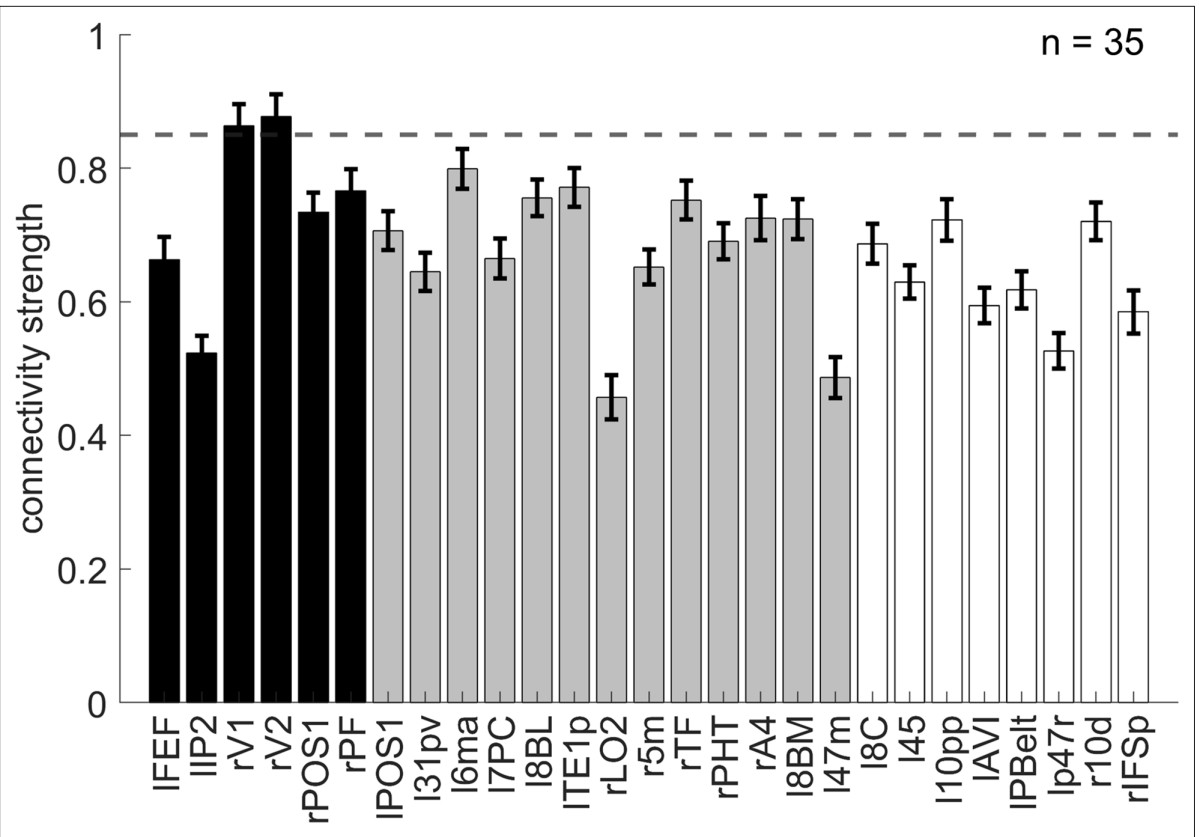

**Figure 8.** The representational connectivity between the right 8C area and the cortical regions showing significant encoding of orientation. The black bars represent regions showing both the overall orientation effect and higher encoding strength of orientation in incongruent than congruent conditions; the gray bars are regions showing only the overall orientation effect but not higher encoding strength of orientation in incongruent than congruent conditions; and the white bars are regions not showing any of the effects of interest (i.e., uncorrected p>0.3 for all the conflict type, orientation, congruency, target, response, spatial Stroop distractor, and Simon distractor effects). Regions plotted in gray and white bars serve as controlled baseline. Error bars are the standard error of the mean. The dashed line indicates the upper bound of the 95% confidence interval of the highest connectivity of controlled regions (i.e., left 6ma). l = left, r = right.

by a limited set of cognitive control processes. In this study, we hypothesized that this issue can be addressed if conflicts are organized as a cognitive space. Leveraging the well-known dissociation between the spatial Stroop and Simon conflicts (*Li et al., 2014*; *Liu et al., 2010*; *Wang et al., 2014*), we designed five conflict types that are systematically different from each other. The cognitive space hypothesis predicted that the representational proximity/distance between two conflict types scales with their similarities/dissimilarities, which was tested at both behavioral and neural levels. Behaviorally, we found that the CSEs were linearly modulated by conflict similarity between consecutive trials, replicating and extending our previous study (*Yang et al., 2021*). BOLD activity patterns in the right dlPFC further showed that the representational similarity between conflict types was modulated by their conflict similarity, and that strength of the modulation was positively associated with the modulation of conflict similarity on the behavioral CSE. We also observed that activity in two brain regions (right IPS and right dlPFC) was parametrically modulated by the conflict type difference, though they did not directly explain the behavioral results. Additionally, we found that the visual regions encoded the spatial orientation of the stimulus location, which might provide the essential concrete information to determine the conflict type. Together, these results suggest that conflicts may be organized in a cognitive space that enables a limited set of cognitive control processes to resolve a wide variety of distinct types of conflicts.

Conventionally, the domain-general view of control suggests a common representation for different types of conflicts (*Figure 9*, left), while the domain-specific view suggests dissociated representations for different types (*Figure 9*, right). Previous research on this topic often adopts a binary manipulation of conflicts (*Braem et al., 2014*) (i.e., each domain only has one conflict type) and gathered

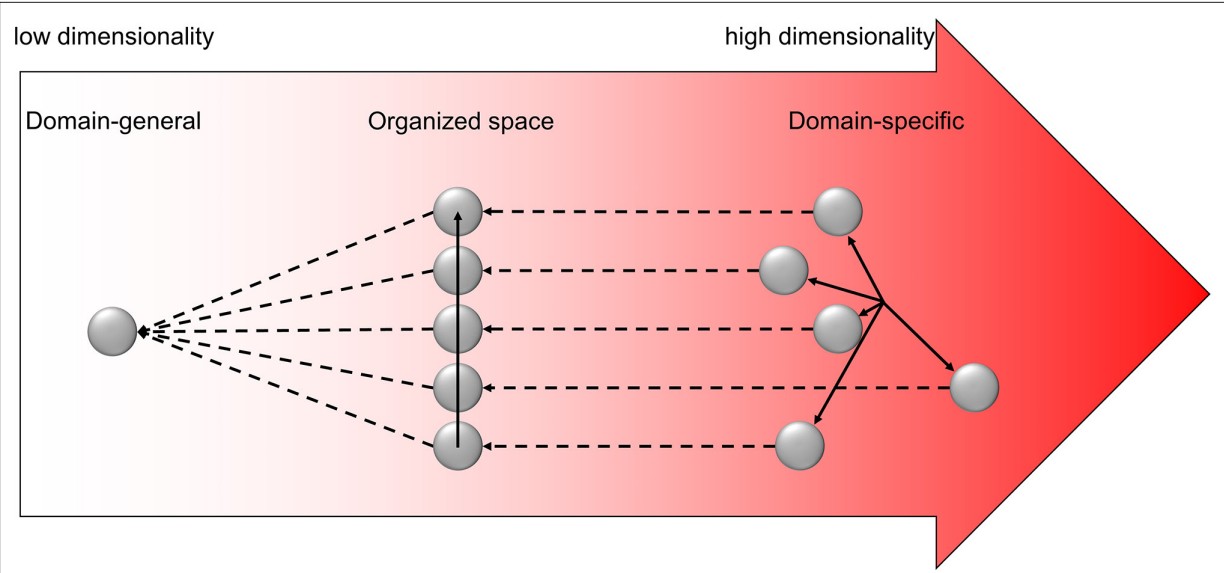

**Figure 9.** Illustration of the hypothesized dimensionalities of different representations. The shade of the red color indicates the degree of dimensionality (i.e., how many dimensions are needed to represent different states). The dimensionality of domain-general representation is extremely low, with all representations compressed to one dot. The dimensionality of domain-specific representation is extremely high, with each control state encoded in a unique and orthogonal dimension. The dimensionality of the organized representation is modest, enabling distant states to be separated but also allowing close states to share representations. The solid arrows show the axes of different dimensions. The dashed arrows indicate how the representational dimensionality can be reduced by projecting the independent dimensions to a common dimension.

evidence for the domain-general/-specific view with the presence/absence of CSE, respectively. Here, we parametrically manipulated the similarity of conflict types and found the CSE systematically vary with conflict similarity level, demonstrating that cognitive control is neither purely domain-general nor purely domain-specific, but can be reconciled as a cognitive space (*Bellmund et al., 2018*; *Figure 9*, middle). The model comparison analysis also showed that the cognitive-space model explained the representation in right DLPFC better than the domain-general or domain-specific models. Specifically, the cognitive space provides a solution to use a single cognitive space organization to encode different types of conflicts that are close (domain-general) or distant (domain-specific) to each other. It also shows the potential for how various conflict types can be coded using limited resources (i.e., as different points in a low-dimensional cognitive space), as suggested by its out-of-sample predictability. Moreover, geometry can also emerge in the cognitive space (*Fu et al., 2022*), which will allow for decomposition of a conflict type (e.g., how much conflict in each of the dimensions in the cognitive space) so that it can be mapped into the limited set of cognitive control processes. Such geometry enables fast learning of cognitive control settings from similar conflict types by providing a measure of similarity (e.g., as distance in space).

If the dimensionality of the cognitive space of conflict is extremely high, the cognitive space solution would suffer the same criticism as the domain-specificity theory. We argue that the dimensionality is manageable for the human brain as task information unrelated to differentiating conflicts can be removed. For example, the Simon conflict can be represented in a space consisting of spatial location, stimulus information, and responses. Thus, the dimensionality of the cognitive space of conflicts should not exceed the number of represented features. The dimensionality can be further reduced as humans selectively represent a small number of features when learning task representations (e.g., spatial information is reduced to the horizontal dimension from the 3D space we live in) (*Niv, 2019*). Consistently, we observed a low dimensional (1.19D) space representing the five conflict types. This is expected since the only manipulated variable is the angular distance between conflict types. The reduced dimensionality does not only require less effort to represent the conflict, but also facilitates generalization of cognitive control settings among different conflict types (*Badre et al., 2021*).

Although our finding of cognitive space in the right dlPFC differs from other cognitive space studies (*Constantinescu et al., 2016*; *Park et al., 2020*; *Schuck et al., 2016*) that highlighted the orbitofrontal

and hippocampus regions, it is consistent with the cognitive control literature. The prefrontal cortex has long been believed to be a key region of cognitive control representation (*Mansouri et al., 2007*; *Miller and Cohen, 2001*; *Milner, 1963*) and is widely engaged in multiple task demands (*Cole et al., 2013*; *Duncan, 2010*). However, it is not until recently that the multivariate representation in this region has been examined. For instance, *Vaidya et al., 2021* reported that frontal regions represented latent states that are organized hierarchically. *Freund et al., 2021a* showed that dlPFC encoded the target and congruency in a typical color-word Stroop task. Taken together, we suggest that the right dlPFC might flexibly encode a variety of cognitive spaces to meet the dynamic task demands. In addition, we found no such representation in the left dlPFC, indicating a possible lateralization. Previous studies showed that the left dlPFC was related to the expectancy-related attentional set upregulation, while the right dlPFC was related to the online adjustment of control (*Friehs et al., 2020*; *Vanderhasselt et al., 2009*), which is consistent with our findings. Moreover, the right PFC also represents a composition of single rules (*Reverberi et al., 2012*), which may explain how the spatial Stroop and Simon types can be jointly encoded in a single space.

Previous researchers have proposed an 'expected value of control (EVC)' theory, which posits that the brain can evaluate the cost and benefit associated with executing control for a demanding task, such as the conflict task, and specify the optimal control strength (*Shenhav et al., 2013*). For instance, *Grahek et al., 2022* found that more frequently switching goals when doing a Stroop task were achieved by adjusting smaller control intensity. Our work complements the EVC theory by further investigating the neural representation of different conflict conditions and how these representations can be evaluated to facilitate conflict resolution. We found that different conflict conditions could be efficiently represented in a cognitive space encoded by the right dlPFC, and participants with stronger cognitive space representation have also adjusted their conflict control to a greater extent based on the conflict similarity (*Figure 6C*). The finding suggests that the cognitive space organization of conflicts guides cognitive control to adjust behavior. Previous studies have shown that participants may adopt different strategies to represent a task, with the model-based strategies benefitting goal-related behaviors more than the model-free strategies (*Rmus et al., 2022*). Similarly, we propose that cognitive space could serve as a mental model to assist fast learning and efficient organization of cognitive control settings. Specifically, the cognitive space representation may provide a principle for how our brain evaluates the expected cost of switching and the benefit of generalization between states and selects the path with the best cost–benefit tradeoff (*Abrahamse et al., 2016*; *Shenhav et al., 2013*). The proximity between two states in cognitive space could reflect both the expected cognitive demand required to transition and the useful mechanisms to adapt from. The closer the two conditions are in cognitive space, the lower the expected switching cost and the higher the generalizability when transitioning between them. With the organization of a cognitive space, a new conflict can be quickly assigned a location in the cognitive space, which will facilitate the development of cognitive control settings for this conflict by interpolating nearby conflicts and/or projecting the location to axes representing different cognitive control processes, thus leading to a stronger CSE when following a more similar conflict condition. On the other hand, without a cognitive space, there would be no measure of similarity between conflicts on different trials, hence limiting the ability of fast learning of cognitive control setting from similar trials.

The cognitive space in the right dlPFC appears to be an abstraction of concrete information from the visual regions. We found that the right V1 and V2 encoded the spatial orientation of the target location (*Figure 7*) and showed strong representational connectivity with the right dlPFC (*Figure 8*), suggesting that there might be information exchange between these regions. We speculate that the representation of spatial orientation may have provided the essential perceptual information to determine the conflict type (*Figure 1*) and thus served as the critical input for the cognitive space. The conflict-type representation further incorporates the stimulus–response mapping rules to the spatial orientation representation, so that vertically symmetric orientations can be recognized as the same conflict type (*Figure 4*). In other words, the representation of conflict type involves the compression of perceptual information (*Flesch et al., 2022*), which is consistent with the idea of a low-dimensional representation of cognitive control (*Badre et al., 2021*; *MacDowell et al., 2022*). The compression and abstraction processes might be why the frontoparietal regions are the top of hierarchy of information processing (*Gilbert and Li, 2013*) and why the frontoparietal regions are widely engaged in multiple task demands (*Duncan, 2013*).

Although the spatial orientation information in our design could be helpful to the construction of cognitive space, the cognitive space itself was independent of the stimulus-level representation of the task. We found the conflict similarity modulation on CSE did not change with more training, indicating that the cognitive space did not depend on strategies that could be learned through training. Instead, the cognitive space should be determined by the intrinsic similarity structure of the task design. For example, a previous study (*Freitas and Clark, 2015*) has found that the CSE across different versions of spatial Stroop and flanker tasks was stronger than that across either of the two conflicts and Simon. In their designs, the stimulus similarity was controlled at the same level, so the difference in CSE was only attributable to the similar dimensional overlap between Stroop and flanker tasks, in contrast to the Simon task. Furthermore, recent studies showed that the cognitive space generally exists to represent structured latent states (e.g., *Vaidya and Badre, 2022*), mental strategy cost (*Grahek et al., 2022*), and social hierarchies (*Park et al., 2020*). Therefore, cognitive space is likely a universal strategy that can be applied to different scenarios.

With conventional univariate analyses, we observed that the overall congruency effect was located at the medial frontal region (i.e., pre-SMA), which is consistent with previous studies (*Botvinick et al., 2004*; *Fu et al., 2022*). Beyond that, we also found regions that can be parametrically modulated by conflict-type difference, including right IPS and right dlPFC (modulated by Simon difference). The right lateralization of these regions is consistent with a previous finding (*Li et al., 2017*). The parametric encoding of conflict also mirrors prior research showing the parametric encoding of task demands (*Dagher et al., 1999*; *Ritz and Shenhav, 2023*). The scaling of brain activities based on conflict difference is potentially important to the representational organization of different types of conflicts. However, we did not observe their brain–behavioral relevance. One possible reason is that the conflict (dis)similarity is a combination of (dis)similarity of spatial Stroop and Simon conflicts, but each univariate region only reflects difference along a single conflict domain. Also likely, the representational geometry is more of a multivariate problem than what univariate activities can capture (*Freund et al., 2021b*). Future studies may adopt approaches such as repetition suppression-induced fMRI adaptation (*Badre et al., 2021*) to test the role of univariate activities in task representations.

Recently an interesting debate has arisen concerning whether cognitive control should be considered as a process or a representation (*Freund et al., 2021b*). Traditionally, cognitive control has been predominantly viewed as a process. However, the study of its representation has gained more and more attention. While it may not be as straightforward as the visual representation (e.g., creating a mental image from a real image in the visual area), cognitive control can have its own form of representation. An influential theory, *Marr, 1982* three-level model proposed that representation serves as the algorithm of the process to achieve a goal based on the input. In other words, representation can encompass a dynamic process rather than being limited to static stimuli. Building on this perspective, we posit that the representation of cognitive control consists of an array of dynamic representations embedded within the overall process. A similar idea has been proposed that the representation of task profiles can be progressively constructed with time in the brain (*Kikumoto and Mayr, 2020*). Moreover, we anticipate that the representation of cognitive space is most prominently involved at two critical stages to guide the transference of behavioral CSE. The first stage involves the evaluation of control demands, where the representational distance/similarity between previous and current trials influences the adjustment of cognitive control. The second stage pertains to control execution, where the switch from one control state to another follows a path within the cognitive space. However, we were unable to fully distinguish between these two stages due to the low temporal resolution of fMRI signals in our study. Future research seeking to delve deeper into this question may benefit from methodologies with higher temporal resolutions, such as EEG and MEG.

Several interesting questions remains to be answered. For example, is the dimension of the unified space across conflict-inducing tasks solely determined by the number of conflict sources? Is this unified space adaptively adjusted within the same brain region? Can we effectively map any sources of conflict with completely different stimuli into a single space? Does the cognitive space vary from population to population, such as between the normal people and patients?

## Methodological implications

Previous studies with mixed conflicts have applied mainly categorical manipulations of conflict types, such as the multi-source interference task (*Fu et al., 2022*) and color Stroop–Simon task (*Liu et al.,*

*2010*). The categorical manipulations make it difficult to quantify conceptual similarity between conflict types and hence limit the ability to test whether neural representations of conflict capture conceptual similarity. To the best of our knowledge, no previous studies have manipulated the conflict types parametrically. This gap highlights a broader challenge within cognitive science: effectively manipulating and measuring similarity levels for conflicts, as well as other high-level cognitive processes, which are inherently abstract. The use of an experimental paradigm that permits parametric manipulation of conflict similarity provides a way to systematically investigate the organization of cognitive control, as well as its influence on adaptive behaviors. Moreover, the cross-subject RSA provides high sensitivity to the variables of interest and the ability to separate confounding factors. For instance, in addition to dissociating conflict type from orientation, we dissociated target from response, and spatial Stroop distractor from Simon distractor. We further showed cognitive control can both enhance the target representation and suppress the distractor representation (*Figure 5E*), which is in line with previous studies (*Polk et al., 2008*; *Ritz and Shenhav, 2022*).

## Limitations

A few limitations of this study need to be noted. To parametrically manipulate the conflict similarity levels, we adopted the spatial Stroop–Simon paradigm that enables parametrical combinations of spatial Stroop and Simon conflicts. However, since this paradigm is a two-alternative forced choice design, the behavioral CSE is not a pure measure of adjusted control but could be partly confounded by bottom-up factors such as feature integration (*Hommel et al., 2004*). Future studies may replicate our findings with a multiple-choice design (including more varied stimulus sets, locations, and responses) with confound-free trial sequences (*Braem et al., 2019*). Another limitation is that in our design, the spatial Stroop and Simon effects are highly anticorrelated. This constraint may make the five conflict types represented in a unidimensional space (e.g., a circle) embedded in a 2D space. This limitation also means we cannot conclusively rule out the possibility of a real unidimensional space driven solely by spatial Stroop or Simon conflicts. However, this appears unlikely as it would imply that our manipulation of conflict types merely represented varying levels of a single conflict, akin to manipulating task difficulty when everything else being equal. If task difficulty were the primary variable, we would expect to see greater representational similarity between task conditions of similar difficulty, such as the Stroop and Simon conditions, which demonstrates comparable congruency effects (see *Figure 2—figure supplement 1*). Contrary to this, our findings reveal that the Stroop-only and Simon-only conditions exhibit the lowest representational similarity (*Figure 6—figure supplement 1*). Furthermore, *Fu et al., 2022* have shown that the representation of mixtures of Simon and Flanker conflicts was compositional, rather than reflecting single dimension, which also applies to our cases. Future studies may test the 2D cognitive space with fully independent conditions. A possible improvement to our current design would be to include left, right, up, and down arrows represented in a grid formation across four spatially separate quadrants, with each arrow mapped to its own response button. Additionally, our study is not a comprehensive test of the cognitive space hypothesis but aimed primarily to provide original evidence for the geometry of cognitive space in representing conflict information in cognitive control. Future research should examine other aspects of the cognitive space such as its dimensionality, its applicability to other conflict tasks such as Eriksen Flanker task, and its relevance to other cognitive abilities, such as cognitive flexibility and learning.

In sum, we showed that the cognitive control can be parametrically encoded in the right dlPFC and guides cognitive control to adjust goal-directed behavior. This finding suggests that different cognitive control states may be encoded in an abstract cognitive space, which reconciles the long-standing debate between the domain-general and domain-specific views of cognitive control and provides a parsimonious and more broadly applicable framework for understanding how our brains efficiently and flexibly represents multiple task settings.

## Materials and methods
### Subjects

In experiment 1, we enrolled 33 college students (ages 19–28 y, average 21.5 ± 2.3 y; 19 males). In experiment 2, 36 college students were recruited, one of which was excluded due to not following task instructions. The final sample of experiment 2 consisted of 35 participants (ages 19–29 y, average

22.3 ± 2.5 y; 17 males). The sample sizes were determined based on our previous study (*Yang et al., 2021*). All participants reported no history of psychiatric or neurological disorders and were right-handed, with normal or corrected-to-normal vision. The experiments were approved by the Institutional Review Board of the Institute of Psychology, Chinese Academy of Science (approval #H19036). Informed consent was obtained from all subjects.

## Method details

### Experiment 1

#### Experimental design

We adopted a modified spatial Stroop–Simon task (*Yang et al., 2021*; *Figure 1*). The task was programmed with E-prime 2.0 (Psychological Software Tools, Inc). The stimulus was an upward or downward black arrow (visual angle of ~1°), displayed on a 17-inch LCD monitor with a viewing distance of ~60 cm. The arrow appeared inside a gray square at 1 of 10 locations with the same distance from the center of the screen, including two horizontal (left and right), two vertical (top and bottom), and six corner (orientations of 22.5°, 45°, and 67.5°) locations. The distance from the arrow to the screen center was approximately 3°. To dissociate orientation of stimulus locations and conflict types (see below), participants were randomly assigned to two sets of stimulus locations (one included top-right and bottom-left quadrants, and the other included top-left and bottom-right quadrants).

Each trial started with a fixation cross displayed in the center for 100–300 ms, followed by the arrow for 600 ms and another fixation cross for 1100–1300 ms (the total trial length was fixed at 2000 ms). Participants were instructed to respond to the pointing direction of the arrow by pressing a left or right button and to ignore its location. The mapping between the arrow orientations and the response buttons was counterbalanced across participants. The task design introduced two possible sources of conflicts: on one hand, the direction of the arrow is either congruent or incongruent with the vertical location of the arrow, thus introducing a spatial Stroop conflict (*Lu and Proctor, 1995*; *MacLeod, 1991*), which contains the dimensional overlap between task-relevant stimulus and task-irrelevant stimulus (*Kornblum et al., 1990*) on the other hand, the response (left or right button) is either congruent or incongruent with the horizontal location of the arrow, thus introducing a Simon conflict (*Lu and Proctor, 1995*; *Simon and Small, 1969*), which contains the dimensional overlap between task-irrelevant stimulus and response (*Kornblum et al., 1990*). Therefore, the five polar orientations of the stimulus location (from 0 to 90°) defined five unique combinations of spatial Stroop and Simon conflicts, with more similar orientations having more similar composition of conflicts. More generally, the spatial orientation of the arrow location relative to the center of the screen forms a cognitive space of different blending of spatial Stroop and Simon conflicts.

The formal task consisted of 30 runs of 101 trials each, divided into three sessions of 10 runs each. The participants completed one session each time and all three sessions within 1 wk. Before each session, the participants performed training blocks of 20 trials repeatedly until the accuracy reached 90% in the most recent block. The trial sequences of the formal task were pseudo-randomly generated to ensure that each of the task conditions and their transitions occurred with equal number of trials.

### Experiment 2

#### Experimental design

The apparatus, stimuli, and procedure were identical to experiment 1 except for the changes below. The stimuli were back projected onto a screen (with viewing angle being ~3.9° between the arrow and the center of the screen) behind the subject and viewed via a surface mirror mounted onto the head coil. Due to the time constraints of fMRI scanning, the trial numbers decreased to a total of 340, divided into two runs with 170 trials each. To obtain a better hemodynamic model fitting, we generated two pseudo-random sequences optimized with a genetic algorithm (*Wager and Nichols, 2003*) conducted by the NeuroDesign package (*Durnez et al., 2018*). In detail, two sequences of 170 trials each were generated independently with the NeuroDesign package (*Durnez et al., 2018*). Each sequence was initialized as 10 consecutive sub-blocks of each condition (incongruent and congruent) for each conflict type (Stroop, $St_HSm_L$, $St_MSm_M$, $St_LSm_H$, and Simon). The contrasts of interest were the main effect of congruency (i.e., [1 –1 1 –1 1 –1 1 –1 1 –1]) and the parametric effect (i.e., [–2 –2 –1 –1 0 0 1 1 2 2]). The order was optimized after 5000 cycles of crossover, mutation, immigration, fitness, and

natural selection. The final number of trials for different conflict types varied from 64 to 73. In addition, we added 6 s of fixation before each run to allow the stabilization of the hemodynamic signal and 20 s after each run to allow the signal to drop to the baseline.

Before scanning, participants performed two practice sessions. The first one contained 10 trials of center-displayed arrow and the second one contained 32 trials using the same design as the main task. They repeated both sessions until their performance accuracy for each session reached 90%, after which the scanning began.

## fMRI Image acquisition and preprocessing

Functional imaging was performed on a 3T GE scanner (Discovery MR750) using echo-planar imaging (EPI) sensitive to BOLD contrast [in-plane resolution of $3.5 \times 3.5$ mm$^2$, $64 \times 64$ matrix, 37 slices with a thickness of 3.5 mm and no interslice skip, repetition time (TR) of 2000 ms, echo-time (TE) of 30 ms, and a flip angle of 90°]. In addition, a sagittal T1-weighted anatomical image was acquired as a structural reference scan, with a total of 256 slices at a thickness of 1.0 mm with no gap and an in-plane resolution of $1.0 \times 1.0$ mm$^2$.

Results included in this manuscript come from preprocessing performed using fMRIPrep 20.2.0 (RRID:SCR_016216; *Esteban et al., 2019* ), which is based on Nipype 1.5.1 (RRID:SCR_002502; *Gorgolewski et al., 2011*).

### Anatomical data preprocessing

The T1-weighted (T1w) image was corrected for intensity non-uniformity (INU) with N4BiasFieldCorrection (*Tustison et al., 2010*), distributed with ANTs 2.3.3 (RRID:SCR_004757; *Avants et al., 2008*), and used as T1w-reference throughout the workflow. The T1w-reference was then skull-stripped with a Nipype implementation of the antsBrainExtraction.sh workflow (from ANTs), using OASIS30ANTs as target template. Brain tissue segmentation of cerebrospinal fluid (CSF), white matter (WM), and gray matter (GM) was performed on the brain-extracted T1w using fast (FSL 5.0.9, RRID:SCR_002823; *Zhang et al., 2001*). Volume-based spatial normalization to one standard space (MNI152NLin2009cAsym) was performed through nonlinear registration with antsRegistration (ANTs 2.3.3) using brain-extracted versions of both T1w reference and the T1w template. The following template was selected for spatial normalization: ICBM 152 Nonlinear Asymmetrical template version 2009c (RRID:SCR_008796; TemplateFlow ID: MNI152NLin2009cAsym; *Fonov et al., 2009*).

### Functional data preprocessing

Before preprocessing, the first three volumes of the functional images were removed due to the instability of the signal at the beginning of the scan. For each of the five BOLD runs found per subject (across all tasks and sessions), the following preprocessing was performed. First, a reference volume and its skull-stripped version were generated using a custom methodology of fMRIPrep. Susceptibility distortion correction (SDC) was omitted. The BOLD reference was then co-registered to the T1w reference using flirt (FSL 5.0.9; *Jenkinson and Smith, 2001*) with the boundary-based registration (*Greve and Fischl, 2009*) cost function. Co-registration was configured with nine degrees of freedom to account for distortions remaining in the BOLD reference. Head-motion parameters with respect to the BOLD reference (transformation matrices, and six corresponding rotation and translation parameters) are estimated before any spatiotemporal filtering using mcflirt (FSL 5.0.9; *Jenkinson et al., 2002*). BOLD runs were slice-time corrected using 3dTshift from AFNI 20160207 (RRID:SCR_005927; *Jenkinson et al., 2002*). The BOLD time series (including slice-timing correction when applied) were resampled onto their original, native space by applying the transforms to correct for head motion. These resampled BOLD time series will be referred to as preprocessed BOLD in original space or just preprocessed BOLD. The BOLD time series were resampled into standard space, generating a preprocessed BOLD run in MNI152NLin2009cAsym space. First, a reference volume and its skull-stripped version were generated using a custom methodology of fMRIPrep. Several confounding time series were calculated based on the preprocessed BOLD: framewise displacement (FD), DVARS, and three region-wise global signals. FD was computed using two formulations following Power (absolute sum of relative motions) (*Jenkinson et al., 2002*) and Jenkinson (relative root mean square displacement between affines, *Jenkinson and Smith, 2001*). FD and DVARS are calculated for each functional run, both using their implementations in Nipype (following the definitions by *Power et al., 2014* ). The three global signals

are extracted within the CSF, the WM, and the whole-brain masks. Additionally, a set of physiological regressors were extracted to allow for component-based noise correction (CompCor) (*Behzadi et al., 2007*). Principal components are estimated after high-pass filtering the preprocessed BOLD time series (using a discrete cosine filter with 128 s cutoff) for the two CompCor variants: temporal (tCompCor) and anatomical (aCompCor). tCompCor components are then calculated from the top 2% variable voxels within the brain mask. For aCompCor, three probabilistic masks (CSF, WM, and combined CSF + WM) are generated in anatomical space. The implementation differs from that of Behzadi et al. in that instead of eroding the masks by 2 pixels on BOLD space, the aCompCor masks are subtracted a mask of pixels that likely contain a volume fraction of GM. This mask is obtained by thresholding the corresponding partial volume map at 0.05, and it ensures components are not extracted from voxels containing a minimal fraction of GM. Finally, these masks are resampled into BOLD space and binarized by thresholding at 0.99 (as in the original implementation). Components are also calculated separately within the WM and CSF masks. For each CompCor decomposition, the k components with the largest singular values are retained, such that the retained components time series are sufficient to explain 50% of variance across the nuisance mask (CSF, WM, combined, or temporal). The remaining components are dropped from consideration. The head-motion estimates calculated in the correction step were also placed within the corresponding confounds file. The confound time series derived from head-motion estimates and global signals were expanded with the inclusion of temporal derivatives and quadratic terms for each (*Behzadi et al., 2007*). Frames that exceeded a threshold of 0.5 mm FD or 1.5 standardized DVARS were annotated as motion outliers. All resamplings can be performed with a single interpolation step by composing all the pertinent transformations (i.e., head-motion transform matrices, susceptibility distortion correction when available, and co-registrations to anatomical and output spaces). Gridded (volumetric) resamplings were performed using antsApplyTransforms (ANTs), configured with Lanczos interpolation to minimize the smoothing effects of other kernels (*Lanczos, 1964*). Non-gridded (surface) resamplings were performed using mri_vol2surf (FreeSurfer).

Many internal operations of fMRIPrep use Nilearn 0.6.2 (RRID:SCR_001362; *Abraham et al., 2014*), mostly within the functional processing workflow. For more details of the pipeline, see the section corresponding to workflows in fMRIPrep's documentation.

After preprocessing, we resampled the functional data to a spatial resolution of $3 \times 3 \times 3$ mm³. All analyses were conducted in volumetric space, and surface maps are produced with Connectome Workbench for display purpose only.

## Quantification and statistical analysis

### Behavioral analysis

#### Experiment 1

RT and ER were the two dependent variables analyzed. As for RTs, we excluded the first trial of each block (0.9%, for CSE analysis only), error trials (3.8%), trials with RTs beyond 3 SDs or shorter than 200 ms (1.3%), and post-error trials (3.4%). For the ER analysis, the first trial of each block and trials after an error were excluded. To exclude the possible influence of response repetition, we centered the RT and ER data within the response repetition and response alternation conditions separately by replacing condition-specific mean with the global mean for each subject.

To examine the modulation of conflict similarity on the CSE, we organized trials based on a 5 (previous trial conflict type) × 5 (current trial conflict type) × 2 (previous trial congruency) × 2 (current trial congruency) factorial design. As conflict similarity is commutive between conflict types, we expected the previous by current trial conflict-type factorial design to be a symmetrical (e.g., a conflict 1–conflict 2 sequence in theory has the same conflict similarity modulation effect as a conflict 2–conflict 1 sequence), resulting in a total of 15 conditions left for the first two factors of the design (i.e., previous × current trial conflict type). For each previous × current trial conflict-type condition, the conflict similarity between the two trials can be quantified as the cosine of their angular difference. In the current design, there were five possible angular difference levels (0°, 22.5°, 42.5°, 67.5°, and 90°, see *Figure 1C*). We further coded the previous by current trial congruency conditions (hereafter abbreviated as CSE conditions) as CC, CI, IC, and II, with the first and second letters encoding the congruency (C) or incongruency (I) on the previous and current trials, respectively. As the CSE is operationalized as the interaction between previous and current trial congruency, it can be rewritten as a contrast of (CI – CC) – (II – IC). In other words, the load of CSE on CI, CC, II, and IC conditions is

1, –1, –1, and 1, respectively. To estimate the modulation of conflict similarity on the CSE, we built a regressor by calculating the Kronecker product of the conflict similarity scores of the 15 previous × current trial conflict similarity conditions and the CSE loadings of previous × current trial congruency conditions. This regressor was regressed against RT and ER data separately, which were normalized across participants and CSE conditions. The regression was performed using a linear mixed-effect model, with the intercept and the slope of the regressor for the modulation of conflict similarity on the CSE as random effects (across both participants and the four CSE conditions). As a control analysis, we built a similar two-stage model (*Yang et al., 2021*). In the first stage, the CSE [i.e., (CI – CC) – (II – IC)] for each of the previous × current trial conflict similarity condition was computed. In the second stage, CSE was used as the dependent variable and was predicted using conflict similarity across the 15 previous × current trial conflict-type conditions. The regression was also performed using a linear mixed-effect model with the intercept and the slope of the regressor for the modulation of conflict similarity on the CSE as random effects (across participants).

### Experiment 2

Behavioral data was analyzed using the same linear mixed-effect model as experiment 1, with all the CC, CI, IC, and II trials as the dependent variable. In addition, to test whether fMRI activity patterns may explain the behavioral representations differently in congruent and incongruent conditions, we conducted the same analysis to measure behavioral modulation of conflict similarity on the CSE using congruent (CC and IC) and incongruent (CI and II) trials separately.

### Estimation of fMRI activity with univariate general linear model (GLM)

To estimate voxel-wise fMRI activity for each of the experimental conditions, the preprocessed fMRI data of each run were analyzed with the GLM. We conducted three GLMs for different purposes. GLM1 aimed to validate the design of our study by replicating the engagement of frontoparietal activities in conflict processing documented in previous studies (*Jiang and Egner, 2014*; *Li et al., 2017*) and explore the cognitive space-related regions that were parametrically modulated by the conflict type. Preprocessed functional images were smoothed using a 6 mm FWHM Gaussian kernel. We included incongruent and congruent conditions as main regressors and appended a parametric modulator for each condition. The modulation parameters for Stroop, $St_HSm_L$, $St_MSm_M$, $St_LSm_H$, and Simon trials were –2, –1, 0, 1, and 2, respectively. In addition, we also added event-related nuisance regressors, including error/missed trials, outlier trials (slower than 3 SDs of the mean or faster than 200 ms) and trials within two TRs of significant head motion (i.e., outlier TRs, defined as standard DVARS > 1.5 or FD > 0.9 mm from previous TR) (*Jiang et al., 2020*). On average, there were 1.2 outlier TRs for each run. These regressors were convolved with a canonical hemodynamic response function (HRF) in SPM 12 (http://www.fil.ion.ucl.ac.uk/spm). We further added volume-level nuisance regressors, including the six head-motion parameters, the global signal, the white matter signal, the CSF signal, and outlier TRs. Low-frequency signal drifts were filtered using a cutoff period of 128 s. The two runs were regarded as different sessions and incorporated into a single GLM to get more power. This yielded two beta maps (i.e., a main effect map and a parametric modulation map) for the incongruent and congruent conditions, respectively, and for each subject. At the group level, paired *t*-tests were conducted between incongruent and congruent conditions, one for the main effect and the other for the parametric modulation effect. Since the spatial Stroop and Simon conflicts change in the opposite direction to each other, a positive modulation effect would reflect a higher brain activation when there is more Simon conflict, and a negative modulation effect would reflect a higher brain activation for more spatial Stroop conflict. Results were corrected with the probabilistic threshold-free cluster enhancement (pTFCE) and then thresholded by 3dClustSim function in AFNI (*Cox and Hyde, 1997*) with voxel-wise p<0.001 and cluster-wise p<0.05, both one-tailed. To visualize the parametric modulation effects, we conducted a similar GLM (GLM2), except we used incongruent and congruent conditions from each conflict type as separate regressors with no parametric modulation. Then we extracted β-coefficients for each regressor and each participant with regions observed in GLM1 as regions of interest, and finally got the incongruent–congruent contrasts for each conflict type at the individual level. We reported the results in *Figure 3*, *Table 1*, and *Figure 3—figure supplement 1*. Visualization of the univoxel results was made by the MRIcron (https://www.mccauslandcenter.sc.edu/mricro/mricron/).

The GLM3 aimed to prepare for the RSA (see below). There were several differences compared to GLM1. The unsmoothed functional images after preprocessing were used. This model included 20 event-related regressors, one for each of the 5 (conflict type) × 2 (congruency condition) × 2 (arrow direction) conditions. The event-related nuisance regressors were similar to GLM1, but with additional regressors of response repetition and post-error trials to account for the nuisance inter-trial effects. To fully expand the variance, we conducted one GLM analysis for each run. After this procedure, a voxel-wise fMRI activation map was obtained per condition, run and subject.

## Representational similarity analysis

To measure the neural representation of conflict similarity, we adopted the RSA. RSAs were conducted on each of the 360 cortical regions of a volumetric version of the MMP cortical atlas (*Glasser et al., 2016*). To de-correlate the factors of conflict type and orientation of stimulus location, we leveraged the between-subject manipulation of stimulus locations and conducted RSA in a cross-subject fashion (*Figure 4*). Previous studies (e.g., *Chen et al., 2017a*) have demonstrated that consistent multivoxel activation patterns exist across individuals, and successful applications of cross-subject RSA (see review by *Freund et al., 2021b*) and cross-subject decoding approaches *Jiang et al., 2016*; *Tusche et al., 2016* have also been reported. The β estimates from GLM3 were noise-normalized by dividing the original β-coefficients by the square root of the covariance matrix of the error terms (*Nili et al., 2014*). For each cortical region, we calculated the Pearson's correlations between fMRI activity patterns for each run and each subject, yielding a 1400 (20 conditions × 2 runs × 35 participants) × 1400 RSM. The correlations were calculated in a cross-voxel manner using the fMRI activation maps obtained from GLM3 described in the previous section. We excluded within-subject cells from the RSM (thus also excluding the within-run similarity as suggested by *Walther et al., 2016*), and the remaining cells were converted into a vector, which was then z-transformed and submitted to a linear mixed-effect model as the dependent variable. The linear mixed-effect model also included regressors of conflict similarity and orientation similarity. Importantly, conflict similarity was based on how Simon and spatial Stroop conflicts are combined and hence was calculated by first rotating all subject's stimulus location to the top-right and bottom-left quadrants, whereas orientation was calculated using original stimulus locations. As a result, the regressors representing conflict similarity and orientation similarity were de-correlated (*Figure 4A*). Similarity between two conditions was measured as the cosine value of the angular difference. Other regressors included a target similarity regressor (i.e., whether the arrow directions were identical), a response similarity regressor (i.e., whether the correct responses were identical); a spatial Stroop distractor regressor (i.e., vertical distance between two stimulus locations); and a Simon distractor regressor (i.e., horizontal distance between two stimulus locations). Additionally, we also included a regressor denoting the similarity of group (i.e., whether two conditions are within the same subject group, according to the stimulus–response mapping). We also added two regressors including ROI-mean fMRI activations for each condition of the pair to remove the possible univoxel influence on the RSM. A last term was the intercept. To control the artifact due to dependence of the correlation pairs sharing the same subject, we included crossed random effects (i.e., row-wise and column-wise random effects) for the intercept, conflict similarity, orientation, and the group factors (*Chen et al., 2017b*). Individual effects for each regressor were also extracted from the model for brain–behavioral correlation analyses. In brain–behavioral analyses, only the RT was used as behavioral measure to be consistent with the fMRI results, where the error trials were regressed out.

The statistical significance of these β estimates was based on the outputs of the mixed-effect model estimated with the 'fitlme' function in MATLAB 2022a. We adjusted the $t$ and $p$ values with the degrees of freedom calculated through the Satterthwaite approximation method (*Satterthwaite, 1946*). Of note, this approach was applied to all the mixed-effect model analyses in this study. Multiple comparison correction was applied with the Bonferroni approach across all cortical regions at the $p<0.05$ level. To test whether the representation strengths are different between congruent and incongruent conditions, we also conducted the RSA using only congruent (RDM_C) and incongruent (RDM_I) trials separately. The contrast analysis was achieved by an additional model with both RDM_C and RDM_I included, adding the congruency and the interaction between conflict type (and orientation) and congruency as both fixed and random factors. The difference between incongruent and congruent representations was indicated by a significant interaction effect. To visualize the difference,

we plotted the effect-related patterns (the predictor multiplied by the slope, plus the residual) as a function of the similarity levels (*Figure 6D*), and a summary RSM for incongruent and congruent conditions, respectively (*Figure 6—figure supplement 1*).

## Model comparison and representational dimensionality

To estimate whether the right 8C specifically encodes the cognitive space, rather than the domain-general or domain-specific structures, we conducted two more RSAs. We replaced the cognitive space-based conflict similarity matrix in the RSA we reported above (hereafter referred to as the cognitive-space model) with one of the alternative model matrices, with all other regressors equal. The domain-general model treats each conflict type as equivalent, so each two conflict types only differ in the magnitude of their conflict. Therefore, we defined the domain-general matrix as the absolute difference in their congruency effects indexed by the group-averaged RT in experiment 2. Then the z-scored model vector was sign-flipped to reflect similarity instead of distance. The domain-specific model treats each conflict type differently, so we used a diagonal matrix, with within-conflict-type similarities being 1 and all cross-conflict-type similarities being 0.

To better capture the dimensionality of the representational space, we estimated its dimensionality using the participation ratio (*Ito and Murray, 2023*). Since we excluded the within-subject cells from the whole RSM, the whole RSM is an incomplete matrix and could not be used. To resolve this issue, we averaged the cells corresponding to each pair of conflict types to obtain an averaged 5 × 5 RSM matrix, similar to the matrix shown in *Figure 1C*. We then estimated the participation ratio using the formula:

$$\dim = \frac{(\sum_i^m \lambda_i)^2}{\sum_i^m \lambda_i^2},$$

where $\lambda_i$ is the eigenvalue of the RSM and $m$ is the number of eigenvalues.

## Representational connectivity analysis

To explore the possible relevance between the conflict type and the orientation effects, we conducted representational connectivity (*Kriegeskorte et al., 2008*) between regions showing evidence encoding conflict similarity and orientation similarity. We hypothesized that this relationship should exist at the within-subject level, so we conducted this analysis using within-subject RSMs excluding the diagonal. Specifically, the z-transformed RSM vector of each region was extracted and submitted to a mixed linear model, with the RSM of the conflict-type region (i.e., the right 8C) as the dependent variable, and the RSM of one of the orientation regions (e.g., right V2) as the predictor. Intercept and the slope of the regressor were set as random effects at the subject level. The mixed-effect model was conducted for each pair of regions, respectively. Considering there might be strong intrinsic correlations across the RSMs induced by the nuisance factors, such as the within-subject similarity, we added two sets of regions as control. First, we selected regions without showing any effects of interest (i.e., uncorrected ps>0.3 for all the conflict type, orientation, congruency, target, response, spatial Stroop distractor, and Simon distractor effects). Second, we selected regions of orientation effect meeting the first but not the second criterion, to account for the potential correlation between regions of the two partly orthogonal regressors (*Figure 4A*). Regions adjacent to the orientation regions were excluded to avoid the inherent strong similarity they may share. Existence of representational connectivity was defined by a connectivity slope higher than 95% of the standard error above the mean of any control region.

## Acknowledgements

We thank Eliot Hazeltine for valuable comments on a previous version of this article. The work was supported by the National Natural Science Foundation of China and the German Research Foundation (NSFC 62061136001/DFG TRR-169) to XL and China Postdoctoral Science Foundation (2019M650884) to GY.

# Additional information

## Funding

| Funder | Grant reference number | Author |
|---|---|---|
| National Natural Science Foundation of China | NSFC 62061136001 | Xun Liu |
| Deutsche Forschungsgemeinschaft | DFG TRR-169 | Xun Liu |
| China Postdoctoral Science Foundation | 2019M650884 | Guochun Yang |

The funders had no role in study design, data collection and interpretation, or the decision to submit the work for publication.

## Author contributions
Guochun Yang, Conceptualization, Data curation, Formal analysis, Funding acquisition, Validation, Investigation, Visualization, Methodology, Writing - original draft, Project administration, Writing – review and editing; Haiyan Wu, Qi Li, Zhongzheng Fu, Writing – review and editing; Xun Liu, Conceptualization, Supervision, Funding acquisition, Project administration, Writing – review and editing; Jiefeng Jiang, Supervision, Validation, Methodology, Writing – review and editing

## Author ORCIDs
Guochun Yang http://orcid.org/0000-0002-0516-8772
Haiyan Wu http://orcid.org/0000-0001-8869-6636
Xun Liu http://orcid.org/0000-0003-1366-8926
Zhongzheng Fu http://orcid.org/0000-0002-2572-6284

## Ethics
Human subjects: The experiments were approved by the Institutional Review Board of the Institute of Psychology, Chinese Academy of Science (Approval #H19036). Informed consent was obtained from all subjects.

Reviewer #1 (Public Review): https://doi.org/10.7554/eLife.87126.5.sa1
Reviewer #2 (Public Review): https://doi.org/10.7554/eLife.87126.5.sa2
Author Response https://doi.org/10.7554/eLife.87126.5.sa3

# Additional files

## Supplementary files
• MDAR checklist

## Data availability
All data needed to evaluate the conclusions are present in the paper. Raw MRI data (in BIDS format), behavioral data and analysis codes can be accessed at https://osf.io/4b3wd/.

The following dataset was generated:

| Author(s) | Year | Dataset title | Dataset URL | Database and Identifier |
|---|---|---|---|---|
| Yang G, Wu H, Li Q, Liu X, Fu Z, Jiang J | 2022 | Conflicts are represented in a cognitive space to reconcile domain-general and domain-specific cognitive control | https://doi.org/10.17605/OSF.IO/4B3WD | Open Science Framework, 10.17605/OSF.IO/4B3WD |

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
