## [Editor Report · eLife assessment]

Yang et al. investigate whether distinct sources of conflict are represented in a common cognitive space. The study uses an interesting task that mixes different sources of difficulty and reports that the brain appears to represent these sources as a mixture on a continuum in prefrontal areas. While the findings could be **valuable** to theory in this area, there are some concerns with the design and results that raise uncertainty regarding the main conclusion of a shared cognitive space. The authors appropriately acknowledge these limitations while also highlighting the valid contributions that the study makes. Thus, while **solid** evidence is reported here, consistent with the central hypothesis, further experiments are required to support the strictest interpretation.

---

## [Referee Report · Reviewer #1 (Public Review)]

People can perform a wide variety of different tasks, and a long-standing question in cognitive neuroscience is how the properties of different tasks are represented in the brain. The authors develop an interesting task that mixes two different sources of difficulty, and find that the brain appears to represent this mixture on a continuum, in the prefrontal areas involved in resolving task difficulty. While these results are interesting and in several ways compelling, they overlap with previous findings and rely on novel statistical analyses that may require further validation.

Strengths

1. The authors present an interesting and novel task for combining the contributions of stimulus-stimulus and stimulus-response conflict. While this mixture has been measured in the multi-source interference task (MSIT), this task provides a more graded mixture between these two sources of difficulty.

2. The authors do a good job triangulating regions that encoding conflict similarity, looking for the conjunction across several different measures of conflict encoding. These conflict measures use several best-practice approaches towards estimating representational similarity.

3. The authors quantify several salient alternative hypothesis, and systematically distinguish their core results from these alternatives.

4. The question that the authors tackle is important to cognitive control, and they make a solid contribution.

Concerns

1. The framing of 'infinite possible types of conflict' feels like a strawman. While they might be true across stimuli (which may motivate a feature-based account of control), the authors explore the interpolation between two stimuli. Instead, this work provides confirmatory evidence that task difficulty is represented parametrically (e.g., consistent with literatures like n-back, multiple object tracking, and random dot motion). This parametric encoding is standard in feature-based attention, and it's not clear what the cognitive map framing is contributing.

2. The representations within DLPFC appear to treat 100% Stoop and (to a lesser extent) 100% Simon differently than mixed trials. Within mixed trials, the RDM within this region don't strongly match the predictions of the conflict similarity model. It appears that there may be a more complex relationship encoded in this region.

3. To orthogonalized their variables, the authors need to employ a complex linear mixed effects analysis, with a potential influence of implementation details (e.g., high-level interactions and inflated degrees of freedom).

---

## [Referee Report · Reviewer #2 (Public Review)]

Summary

This study examines the construct of "cognitive spaces" as they relate to neural coding schemes present in response conflict tasks. The authors use a novel experimental design in which different types of response conflict (spatial Stroop, Simon) are parametrically manipulated. These conflict types are hypothesized to be encoded jointly, within an abstract "cognitive space", in which distances between task conditions depend only on the similarity of conflict types (i.e., where conditions with similar relative proportions of spatial-Stroop versus Simon conflicts are represented with similar activity patterns). Authors contrast such a representational scheme for conflict with several other conceptually distinct schemes, including a domain-general, domain-specific, and two task-specific schemes. The authors conduct a behavioral and fMRI study to test whether prefrontal cortex activity is correlated to one of these coding schemes. Replicating the authors' prior work, this study demonstrates that sequential behavioral adjustments (the congruency sequence effect) are modulated as a function of the similarity between conflict types. In fMRI data, univariate analyses identified activation in left prefrontal and dorsomedial frontal cortex that was modulated by the amount of Stroop or Simon conflict present, and representational similarity analyses that identified coding of conflict similarity, as predicted under the cognitive space model, in right lateral prefrontal cortex.

Strengths

This study addresses an important question regarding how conflict or difficulty might be encoded in the brain within a computationally efficient representational format. Relative to the other models reported in the paper, the evidence in support of the cognitive space model is solid. The ideas postulated by the authors are interesting and valuable ones, worthy of follow-up work that provides additional necessary scrutiny of the cognitive-space account.

Weaknesses

Future, within-subject experiments will be necessary to disentangle the cognitive space model from confounded task variables. A between-subjects manipulation of stimulus orientation/location renders the results difficult to interpret, as the source and spatial scale of the conflict encoding on cortex may differ from more rigorous (and more typical) within-subject manipulations.

Results are also difficult to interpret because Stroop and Simon conflict are confounded with each other. For interpretability, these two sources of conflict need to be manipulated orthogonally, so that each source of conflict (as well as their interaction) could be separately estimated and compared in terms of neural encoding. For example, it is therefore not clear whether the RSA results are due to encoding of only one type of conflict (Stroop or Simon), to a combination of both, and/or to interactive effects.

Finally, the motivation for the use of the term "cognitive space" to describe results is unclear. Evidence for the mere presence of a graded/parametric neural encoding (i.e., the reported conflict RSA effects) would not seem to be sufficient. Indeed, it is discussed in the manuscript that cognitive spaces/maps allow for flexibility through inference and generalization. Future work should therefore focus on linking neural conflict encoding to inference and generalization more directly.

---

## [Author Response]

The following is the authors’ response to the previous reviews.

**Reviewer #1:**
Concerns Public Review:1)The framing of 'infinite possible types of conflict' feels like a strawman. While they might be true across stimuli (which may motivate a feature-based account of control), the authors explore the interpolation between two stimuli. Instead, this work provides confirmatory evidence that task difficulty is represented parametrically (e.g., consistent with literatures like n-back, multiple object tracking, and random dot motion). This parametric encoding is standard in feature-based attention, and it's not clear what the cognitive map framing is contributing.Suggestion:1. 'infinite combinations'. I'm frankly confused by the authors response. I don't feel like the framing has changed very much, besides a few minor replacements. Previous work in MSIT (e.g., by the author Zhongzheng Fu) has looked at whether conflict levels are represented similarly across conflict types using multivariate analyses. In the paper mentioned by Ritz & Shenhav (2023), the authors looked at whether conflict levels are represented similarly across conflict types using multivariate analyses. It's not clear what this paper contributes theoretically beyond the connections to cognitive maps, which feel like an interpretative framework rather than a testable hypothesis (i.e., these previous paper could have framed their work as cognitive maps).

Response: We acknowledge the limitations inherent in our experimental design, which prevents us from conducting a strict test of the cognitive space view. In our previous revision, we took steps to soften our conclusions and emphasize these limitations. However, we still believe that our study offers valuable and novel insights into the cognitive space, and the tests we conducted are not merely strawman arguments.

Specifically, our study aimed to investigate the fundamental principles of the cognitive space view, as we stated in our manuscript that “the representations of different abstract information are organized continuously and the representational geometry in the cognitive space is determined by the similarity among the represented information (Bellmund et al., 2018)”. While previous research has applied multivariate analyses to understand cognitive control representation, no prior studies had directedly tested the two key hypotheses associated with cognitive space: (1) that cognitive control representation across conflict types is continuous, and (2) that the similarity among representations of different conflict types is determined by their external similarity.

Our study makes a unique contribute by directly testing these properties through a parametric manipulation of different conflict types. This approach differs significantly from previous studies in two ways. First, our parametric manipulation involves more than two levels of conflict similarity, enabling us to directly test the two critical hypotheses mentioned above. Unlike studies such as Fu et al. (2022) and other that have treated different conflict types categorically, we introduced a gradient change in conflict similarity. This differentiation allowed us to employ representational similarity analysis (RSA) over the conflict similarity, which goes beyond mere decoding as utilized in prior work (see more explanation below for the difference between Fu et al., 2022 and our study [1]).

Second, our parametric manipulation of conflict types differs from previous studies that have manipulated task difficulty, and the modulation of multivariate pattern similarity observed in our study could not be attributed by task difficulty. Previous research, including the Ritz & Shenhav (2023) (see below explanation[2]), has primarily shown that task difficulty modulates univoxel brain activation. A recent work by Wen & Egner (2023) reported a gradual change in the multivariate pattern of brain activations across a wide range of frontoparietal areas, supporting the reviewer’s idea that “task difficulty is represented parametrically”. However, we do not believe that our results reflect the task difficulty representation. For instance, in our study, the spatial Stroop-only and Simon-only conditions exhibited similar levels of difficulty, as indicated by their relatively comparable congruency effects (Fig. S1). Despite this similarity in difficulty, we found that the representational similarity between these two conditions was the lowest (see revised Fig. S4, the most off-diagonal value). This observation aligns more closely with our hypothesis that these two conditions are most dissimilar in terms of their conflict types.

[1] Fu et al. (2022) offers important insights into the geometry of cognitive space for conflict processing. They demonstrated that Simon and flanker conflicts could be distinguished by a decoder that leverages the representational geometry within a multidimensional space. However, their model of cognitive space primarily relies on categorical definitions of conflict types (i.e., Simon versus flanker), rather than exploring a parametric manipulation of these conflict types. The categorical manipulations make it difficult to quantify conceptual similarity between conflict types and hence limit the ability to test whether neural representations of conflict capture conceptual similarity. To the best of our knowledge, no previous studies have manipulated the conflict types parametrically. This gap highlights a broader challenge within cognitive science: effectively manipulating and measuring similarity levels for conflicts, as well as other high-level cognitive processes, which are inherently abstract. We therefore believe our parametric manipulation of conflict types, despite its inevitable limitations, is an important contribution to the literature.

We have incorporated the above statements into our revised manuscript: Methodological implications. Previous studies with mixed conflicts have applied mainly categorical manipulations of conflict types, such as the multi-source interference task (Fu et al., 2022) and color Stroop-Simon task (Liu et al., 2010). The categorical manipulations make it difficult to quantify conceptual similarity between conflict types and hence limit the ability to test whether neural representations of conflict capture conceptual similarity. To the best of our knowledge, no previous studies have manipulated the conflict types parametrically. This gap highlights a broader challenge within cognitive science: effectively manipulating and measuring similarity levels for conflicts, as well as other high-level cognitive processes, which are inherently abstract. The use of an experimental paradigm that permits parametric manipulation of conflict similarity provides a way to systematically investigate the organization of cognitive control, as well as its influence on adaptive behaviors.

[2] The work by Ritz & Shenhav (2023) indeed applied multivariate analyses, but they did not test the representational similarity across different levels of task difficulty in a similar way as our investigation into different levels of conflict types, neither did they manipulated conflict types as our study. They first estimated univariate brain activations that were parametrically scaled by task difficulty (e.g., target coherence), yielding one map of parameter estimates (i.e., encoding subspace) for each of the target coherence and distractor congruence. The multivoxel patterns from the above maps were correlated to test whether the target coherence and distractor congruence share the similar neural encoding. It is noteworthy that the encoding of task difficulty in their study is estimated at the univariate level, like the univariate parametric modulation analysis in our study. The representational similarity across target coherence and distractor congruence was the second-order test and did not reflect the similarity across different difficulty levels. Though, we have found another study (Wen & Egner, 2023) that has directly tested the representational similarity across different levels of task difficulty, and they observed a higher representational similarity between conditions with similar difficulty levels within a wide range of brain regions.

Reference:

Wen, T., & Egner, T. (2023). Context-independent scaling of neural responses to task difficulty in the multiple-demand network. Cerebral Cortex, 33(10), 6013-6027. https://doi.org/10.1093/cercor/bhac479

Fu, Z., Beam, D., Chung, J. M., Reed, C. M., Mamelak, A. N., Adolphs, R., & Rutishauser, U. (2022). The geometry of domain-general performance monitoring in the human medial frontal cortex. Science (New York, N.Y.), 376(6593), eabm9922. https://doi.org/10.1126/science.abm9922

Ritz, H., & Shenhav, A. (2023). Orthogonal neural encoding of targets and distractors supports multivariate cognitive control. https://doi.org/10.1101/2022.12.01.518771 Another issue is suggesting mixtures between two types of conflict may be many independent sources of conflict. Again, this feels like the strawman. There's a difference between infinite combinations of stimuli on the one hand, and levels of feature on the other hand. The issue of infinite stimuli is why people have proposed feature-based accounts, which are often parametric, eg color, size, orientation, spatial frequency. Mixing two forms of conflict is interesting, but the task limitations (i.e., highly correlated features) prevent an analysis of whether these are truly mixed (or eg reflect variations on just one of the conflict types). Without being able to compare a mixture between types vs levels of only one type, it's not clear what you can draw from these results re: how these are combined (and not clear how it reconciles the debate between general and specific).

Response: As the reviewer pointed out, a feature (or a parameterization) is an efficient way to encode potentially infinite stimuli. This is the same idea as our hypothesis: different conflict types are represented in a cognitive space akin to concrete features such as a color spectrum. This concept can be illustrated in the figure below.

We would like to clarify that in our study we have manipulated five levels of conflict types, but they all originated from two fundamental sources: vertically spatial Stroop and horizontally Simon conflicts. We agree that the mixture of these two sources does not inherently generate additional conflict sources. However, this mixture does influence the similarity among different conflict conditions, which provides essential variability that is crucial for testing the core hypotheses (i.e., continuity and similarity modulation, see the response above) of the cognitive space view. This clarification is crucial as the reviewer’s impression might have been influenced by our introduction, where we repeatedly emphasized multiple sources of conflicts. Our aim in the introduction was to outline a broader conceptual framework, which might not directly reflect the specific design of our current study. Recognizing the possibility of misinterpretation, we have adjusted our introduction and discussion to place less emphasis on the variety of possible conflict sources. For example, we have removed the expression “The large variety of conflict sources implies that there may be innumerable number of conflict conditions” from the introduction. As we have addressed in the previous response, the observed conflict similarity effect could not be attributed to merely task difficulty. Similarly, the mixture of spatial Stroop and Simon conflicts should not be attributed to one conflict source only; doing so would oversimplify it to an issue of task difficulty, as it would imply that our manipulation of conflict types merely represented varying levels of a single conflict, akin to manipulating task difficulty when everything else being equal. Importantly, the mixed conditions differ from variations along a single conflict source in that they also incorporate components of the other conflict source, thereby introducing difference beyond that would be found within variances of a single conflict source. There are a few additional evidence challenging the single dimension assumption. In our previous revisions, we compared model fittings between the Cognitive-Space model and the Stroop-/Simon-only models, and results showed that the CognitiveSpace model (BIC = 5377093) outperformed the Stroop-Only (BIC = 5377122) and Simon-Only (BIC = 5377096) models. This suggests that mixed conflicts might not be solely reflective of either Stroop or Simon sources, although we did not include these results due to concerns raised by reviewers about the validity of such comparisons, given the high anticorrelation between the two dimensions. Furthermore, Fu et al. (2022) demonstrated that the mixture of Simon and Flanker conflicts (the sf condition) is represented as the vector sum of the Flanker and Simon dimensions within their space model, indicating a compositional nature. Similarly, our mixed conditions are combinations of Stroop and Simon conflicts, and it is plausible that these mixtures represent a fusion of both Stroop and Simon components, rather than just one. Thus, we disagree that the mixture of conflicts is a strawman. In response to this concern, we have included a statement in our limitation section: “Another limitation is that in our design, the spatial Stroop and Simon effects are highly anticorrelated. This constraint may make the five conflict types represented in a unidimensional space (e.g., a circle) embedded in a 2D space. This limitation also means we cannot conclusively rule out the possibility of a real unidimensional space driven solely by spatial Stroop or Simon conflicts. However, this appears unlikely, as it would imply that our manipulation of conflict types merely represented varying levels of a single conflict, akin to manipulating task difficulty when everything else being equal. If task difficulty were the primary variable, we would expect to see greater representational similarity between task conditions of similar difficulty, such as the Stroop and Simon conditions, which demonstrates comparable congruency effects (see Fig. S1). Contrary to this, our findings reveal that the Stroop-only and Simon-only conditions exhibit the lowest representational similarity (Fig. S4). Furthermore, Fu et al. (2022) has shown that the representation of mixtures of Simon and Flanker conflicts was compositional, rather than reflecting single dimension, which also applies to our cases.”

My recommendation would be to dramatically rewrite to reduce the framing of this providing critical evidence in favor of cognitive maps, and being more overt about the limitations of this task. However, the authors are not required to make further revisions in eLife's new model, and it's not clear how my scores would change if they made those revisions (ie the conceptual limitations would remain, the claims would just now match the more limited scope).

Response: With the above rationales and the adjustments we have made in the manuscripts, we believe that we have thoroughly acknowledged and articulated the limitations of our study. Therefore, we have decided against a complete rewrite of the manuscript.

**Public Review:**
1. The representations within DLPFC appear to treat 100% Stoop and (to a lesser extent) 100% Simon differently than mixed trials. Within mixed trials, the RDM within this region don't strongly match the predictions of the conflict similarity model. It appears that there may be a more complex relationship encoded in this region.Suggestion:1. RSMs in the key region of interest. I don't really understand the authors response here either. e.g,. 'It is essential to clarify that our conclusions were based on the significant similarity modulation effect identified in our statistical analysis using the cosine similarity model, where we did not distinguish between the within-Stroop condition and the other four within-conflict conditions (Fig. 7A, now Fig. 8A). This means that the representation of conflict type was not biased by the seemingly disparities in the values shown here'. In Figure 1C, it does look like they are testing this model.It seems like a stronger validation would test just the mixture trials (i.e., ignoring Simon-only and stroop-only). However, simon/stroop-only conditions being qualitatively different does beg the question of whether these are being represented parametrically vs categorically.

Response: We apologize for the confusion caused by our previous response. To clarify, our conclusions have been drawn based on the robust conflict similarity effect.

The conflict similarity regressor is defined by higher values in the diagonal cells (representing within-conflict similarity) than the off-diagonal cells (indicating between-conflict similarity), as illustrated in Fig. 1C and Fig. 8A (now Fig. 4B). It is important to note that this regressor may not be particularly sensitive to the variations within the diagonal cells. Our previous response aimed to emphasize that the inconsistencies observed along the diagonal do not contradict our core hypothesis regarding the conflict similarity effect.

We recognized that since the visualization in Fig. S4, based on the raw RSM (i.e., Pearson correlation), may have been influenced by other regressors in our model than the conflict similarity effect. To reflect pattern similarity with confounding factors controlled for, we have visualized the RSM by including only the fixed effect of the conflict similarity and the residual while excluding all other factors. As shown in the revised Figure S4, the difference between the within-Stroop and other diagonal cells was greatly reduced. Instead, it revealed a clear pattern where that the diagonal values were higher than the off-diagonal values in the incongruent condition, aligning with our hypothesis regarding the conflict similarity modulator. Although some visual distinctions persist within the five diagonal cells (e.g., in the incongruent condition, the Stroop, Simon, and StMSmM conditions appear slightly lower than StHSmL and StLSmM conditions), follow-up one-way ANOVAs among these five diagonal conditions showed no significant differences. This held true for both incongruent and congruent conditions, with Fs < 1. Thus, we conclude that there is no strong evidence supporting the notion that Simon- and spatial Stroop-only conditions are systematically different from other conflict types. As a result, we decided not to exclude these two conflict types from analysis.

**Author response image 2. sa3fig2:** The stronger conflict type similarity effect in incongruent versus congruent conditions. Shown are the summary representational similarity matrices for the right 8C region in incongruent (left) and congruent (right) conditions, respectively. Each cell represents the averaged Pearson correlation (after regressing out all factors except the conflict similarity) of cells with the same conflict type and congruency in the 1400 × 1400 matrix. Note that the seemingly disparities in the values of withinconflict cells (i.e., the diagonal) did not reach significance for either incongruent or congruent trials, Fs < 1.

**Public Review:**
1. To orthogonalized their variables, the authors need to employ a complex linear mixed effects analysis, with a potential influence of implementation details (e.g., high-level interactions and inflated degrees of freedom).Suggestion:1. The DF for a mixed model should not be the number of observations minus the number of fixed effects. The gold standard is to use satterthwaite correction (e.g. in Matlab, fixedEffects(lme,'DFMethod','satterthwaite')), or number of subjects - number of fixed effects (i.e. you want to generalize to new subjects, not just new samples from the same subjects). Honestly, running a 4-way interaction probably is probably using more degrees of freedom than are appropriate given the number of subjects.

Response: We concur with the reviewer’s comment that our previous estimation of degrees of freedom (DFs) was inaccurate. Following your suggestion, we have now applied the “Satterthwaite” approach to approximate the DFs for all our linear mixed effect model analyses. This adjustment has led to the correction of both DFs and p values. In the Methods section, we have mentioned this revision.

“We adjusted the t and p values with the degrees of freedom calculated through the Satterthwaite approximation method (Satterthwaite, 1946). Of note, this approach was applied to all the mixed-effect model analyses in this study.”

The application of this method has indeed resulted in a reduction of our statistical significance. However, our overall conclusions remained robust. Instead of the highly stringent threshold used in our previous version (Bonferonni corrected p < .0001), we have now adopted a relatively more lenient threshold of Bonferonni correction at p < 0.05, which is commonly employed in the literature. Furthermore, it is worth noting that the follow-up criteria 2 and 3 are inherently second-order analyses. Criterion 2 involves examining the interaction effect (conflict similarity effect difference between incongruent and congruent conditions), and criterion 3 involves individual correlation analyses. Due to their second-order nature, these criteria inherently have lower statistical power compared to criterion 1 (Blake & Gangestad, 2020). We thus have applied a more lenient but still typically acceptable false discovery rate (FDR) correction to criteria 2 and 3. This adjustment helps maintain the rigor of our analysis while considering the inherent differences in statistical power across the various criteria. We have mentioned this revision in our manuscript:

“We next tested whether these regions were related to cognitive control by comparing the strength of conflict similarity effect between incongruent and congruent conditions (criterion 2) and correlating the strength to behavioral similarity modulation effect (criterion 3). Given these two criteria pertain to second-order analyses (interaction or individual analyses) and thus might have lower statistical power (Blake & Gangestad, 2020), we applied a more lenient threshold using false discovery rate (FDR) correction (Benjamini & Hochberg, 1995) on the above-mentioned regions.”

With these adjustments, we consistently identified similar brain regions as observed in our previous version. Specifically, we found that only the right 8C region met the three criteria in the conflict similarity analysis. In addition, the regions meeting the criteria for the orientation effect included the FEF and IP2 in left hemisphere, and V1, V2, POS1, and PF in the right hemisphere. We have thoroughly revised the description of our results, updated the figures and tables in both the revised manuscript and supplementary material to accurately reflect these outcomes.

Reference:

Blake, K. R., & Gangestad, S. (2020). On Attenuated Interactions, Measurement Error, and Statistical Power: Guidelines for Social and Personality Psychologists. Pers Soc Psychol Bull, 46(12), 1702-1711. https://doi.org/10.1177/0146167220913363

Minor:1. Figure 8 should come much earlier (e.g, incorporated into Figure 1), and there should be consistent terms for 'cognitive map' and 'conflict similarity'.

Response: We appreciate this suggestion. Considering that Figure 7 (“The crosssubject RSA model and the rationale”) also describes the models, we have merged Figure 7 and 8 and moved the new figure ahead, before we report the RSA results. Now you could find it in the new Figure 4, see below. We did not incorporate them into Figure 1 since Figure 1 is already too crowded.

**Author response image 3. sa3fig3:** Fig 4. Rationale of the cross-subject RSA model and the schematic of key RSMs. (A) The RSM is calculated as the Pearson’s correlation between each pair of conditions across the 35 subjects. For 17 subjects, the stimuli were displayed on the top-left and bottom-right quadrants, and they were asked to respond with left hand to the upward arrow and right hand to the downward arrow. For the other 18 subjects, the stimuli were displayed on the top-right and bottom-left quadrants, and they were asked to respond with left hand to the downward arrow and right hand to the upward arrow. Within each subject, the conflict type and orientation regressors were perfectly covaried. For instance, the same conflict type will always be on the same orientation. To de-correlate conflict type and orientation effects, we conducted the RSA across subjects from different groups. For example, the bottom-right panel highlights the example conditions that are orthogonal to each other on the orientation, response, and Simon distractor, whereas their conflict type, target and spatial Stroop distractor are the same. The dashed boxes show the possible target locations for different conditions. (B) and (C) show the orthogonality between conflict similarity and orientation RSMs. The within-subject RSMs (e.g., Group1-Group1) for conflict similarity and orientation are all the same, but the cross-group correlations (e.g., Group2-Group1) are different. Therefore, we can separate the contribution of these two effects when including them as different regressors in the same linear regression model. (D) and (E) show the two alternative models. Like the cosine model (B), within-group trial pairs resemble betweengroup trial pairs in these two models. The domain-specific model is an identity matrix. The domaingeneral model is estimated from the absolute difference of behavioral congruency effect, but scaled to 0 (lowest similarity) – 1 (highest similarity) to aid comparison. The plotted matrices in B-E include only one subject each from Group 1 and Group 2. Numbers 1-5 indicate the conflict type conditions, for spatial Stroop, StHSmL, StMSmM, StLSmH, and Simon, respectively. The thin lines separate four different sub-conditions, i.e., target arrow (up, down) × congruency (incongruent, congruent), within each conflict type.

In our manuscript, the term “cognitive map/space” was used when explaining the results in a theoretical perspective, whereas the “conflict similarity” was used to describe the regressor within the RSA. These terms serve distinct purposes in our study and cannot be interchangeably substituted. Therefore, we have retained them in their current format. However, we recognize that the initial introduction of the “Cognitive-Space model” may have appeared somewhat abrupt. To address this, we have included a brief explanatory note: “The model described above employs the cosine similarity measure to define conflict similarity and will be referred to as the Cognitive-Space model.”